# Structures of transcription preinitiation complex engaged with the +1 nucleosome

Haibo Wang[1,4], Sandra Schilbach[1], Momchil Ninov[2], Henning Urlaub[2,3] & Patrick Cramer [1]✉

The preinitiation complex (PIC) assembles on promoters of protein-coding genes to position RNA polymerase II (Pol II) for transcription initiation. Previous structural studies revealed the PIC on different promoters, but did not address how the PIC assembles within chromatin. In the yeast *Saccharomyces cerevisiae*, PIC assembly occurs adjacent to the +1 nucleosome that is located downstream of the core promoter. Here we present cryo-EM structures of the yeast PIC bound to promoter DNA and the +1 nucleosome located at three different positions. The general transcription factor TFIIH engages with the incoming downstream nucleosome and its translocase subunit Ssl2 (XPB in human TFIIH) drives the rotation of the +1 nucleosome leading to partial detachment of nucleosomal DNA and intimate interactions between TFIIH and the nucleosome. The structures provide insights into how transcription initiation can be influenced by the +1 nucleosome and may explain why the transcription start site is often located roughly 60 base pairs upstream of the dyad of the +1 nucleosome in yeast.

Previous studies provided structures of yeast and human preinitiation complexes (PICs) on various promoters[1–9]. However, promoters are flanked by nucleosomes within the chromatin environment in vivo[10,11] and thus PIC structures must also be determined in the presence of nucleosomes. In the yeast *Saccharomyces cerevisiae*, PIC assembly occurs adjacent to the +1 nucleosome[12,13], which resides at the downstream end of the core promoter. The +1 nucleosome is often well-positioned[14,15] and thought to be involved in PIC assembly[13], although the underlying mechanisms are unclear[16]. The +1 nucleosome is associated with the PIC at most Pol II promoters[17]. Correct positioning of the +1 nucleosome positively influences binding of the TATA box-binding protein (TBP), selection of the transcription start site (TSS) and transcription activity[18–21] in vivo.

Here we use a combination of biochemistry and cryo-electron microscopy (cryo-EM) to provide structural insights into the PIC in the context of the +1 nucleosome. We show that TFIIH can engage with the +1 nucleosome in different ways and provide evidence that there

is a preferred mode of TFIIH-nucleosome interaction that relies on multiple contacts. Finally, we use the PIC–nucleosome structures and modeling to provide a molecular explanation for long-standing observations on the preferred relative position of the TSS and the location of the +1 nucleosome. Our work thus provides the basis for a detailed analysis of structural and functional PIC-nucleosome interactions at gene promoters.

## Results

### Formation of PIC–nucleosome complex

To investigate PIC assembly in the presence of the +1 nucleosome, we formed a PIC from the yeast *S. cerevisiae* on a promoter flanked by a +1 nucleosome (Fig. 1a,b). At most yeast promoters, the +1 nucleosome covers the TSS, which is typically located roughly 10–15 base pairs (bp) downstream of the proximal border of the nucleosome[15,20,22]. To mimic this natural arrangement, we prepared a *His4* promoter template with a Widom-601-derived nucleosome positioning sequence that places

[1]Department of Molecular Biology, Max Planck Institute for Multidisciplinary Sciences, Göttingen, Germany. [2]Bioanalytical Mass Spectrometry, Max Planck Institute for Multidisciplinary Sciences, Göttingen, Germany. [3]Institute of Clinical Chemistry, Bioanalytics Group, University Medical Center Göttingen, Göttingen, Germany. [4]Present address: Cancer Institute of the Second Affiliated Hospital and Institute of Translational Medicine, Zhejiang University School of Medicine, Hangzhou, China. ✉e-mail: patrick.cramer@mpinat.mpg.de

the +1 nucleosome at a position in which the TSS is 10 bp downstream of the proximal border of the nucleosome (Fig. 1a). We reconstituted a nucleosome on this DNA and used the obtained nucleosomal template for in vitro assembly of the PIC[4] (Extended Data Fig. 1a and Methods).

## The +1 nucleosome represses transcription

To evaluate the effect of the +1 nucleosome on transcription activity, we performed promoter-dependent in vitro transcription assays[9] (Fig. 1b and Methods). We found that the presence of the +1 nucleosome strongly reduced RNA synthesis in this assay (Fig. 1c). The amount of full-length RNA product was reduced to 20% and shorter RNA transcripts were produced, as expected from Pol II stalling within the nucleosome[23] (Fig. 1c). To test whether the reduction of RNA synthesis is caused by the high stability of a nucleosome obtained on a Widom-601 derived positioning sequence, we repeated the assay with a template containing the natural *His4* promoter sequence. We observed that the level of RNA synthesis was again repressed, albeit to a lesser extent than for the original template containing the nucleosome positioning sequence (Extended Data Fig. 1b). These results indicate that the degree of transcription reduction is related to the stability of the nucleosome while any type of nucleosome causes a decrease in RNA production.

## Cryo-EM structure determination

We then determined cryo-EM structures of the reconstituted PIC–nucleosome complex (Methods) in the absence (complex A) or presence (complex B) of nucleoside triphosphates (NTPs) under the conditions of our transcription assay. Classification of the data identified a subset of particles that contained the complete complex (Extended Data Figs. 2 and 3). Cryo-EM densities for TFIIH and the nucleosome were further improved by focused refinement. For complex A we obtained a reconstruction at an overall resolution of 3.3 Å, with local resolutions of 2.9 Å for Pol II, 3.2 Å for the nucleosome and 3.7 Å for TFIIH (Extended Data Figs. 2 and 4 and Supplementary Video 1). Complex B was resolved at an overall resolution of 4.0 Å, with local resolutions of 3.4 Å for Pol II, 3.6 Å for the nucleosome and 3.9 Å for TFIIH (Extended Data Figs. 3 and 4 and Supplementary Video 1). The structures were obtained with the use of atomic models of the PIC[9] and the nucleosome[24] and subsequent manual modeling, leading to good stereochemistry (Table 1).

## PIC structures are largely unchanged

In both cryo-EM structures, the overall conformation of the PIC resembles that in the absence of the nucleosome[4,9], except for a minor rotation of TFIIH with respect to the rest of the PIC (Fig. 2 and Extended Data Fig. 5a). The complex adopts the previously described closed promoter state with distorted DNA[25] and shows the TFIIH ATPase Ssl2 in the pre-translocation state. The DNA is located above the active center cleft and the initially melting DNA region is flanked by the Rpb1 clamp head loop and the TFIIF charged region as observed before[9] (Fig. 2a,b). In contrast to the closed promoter state observed here, previous cryo-EM studies of the yeast PIC revealed a large portion of PIC particles in the open promoter state[4,9], suggesting that the +1 nucleosome counteracts DNA opening and that impaired DNA opening is responsible for the observed suppression of transcription in the presence of the +1 nucleosome.

## Rotation of the +1 nucleosome

Comparison of the structures of complexes A and B shows that the +1 nucleosome is rotated by roughly 75° in the presence of NTPs (Fig. 3a). This rotation is apparently caused by the translocase activity of TFIIH subunit Ssl2 that hydrolyzes ATP to propel downstream DNA into the PIC. Such ATPase action is predicted to cause a rotation of the nucleosome by 30–40° with respect to the PIC for each translocated DNA base pair. The observed roughly 75° rotation would thus correspond to a DNA translocation of 2 bp toward the active center of Pol II, and this is reflected by an observed bending of the DNA duplex into the cleft of Pol II around the initially melted region (Extended Data Fig. 5b). The

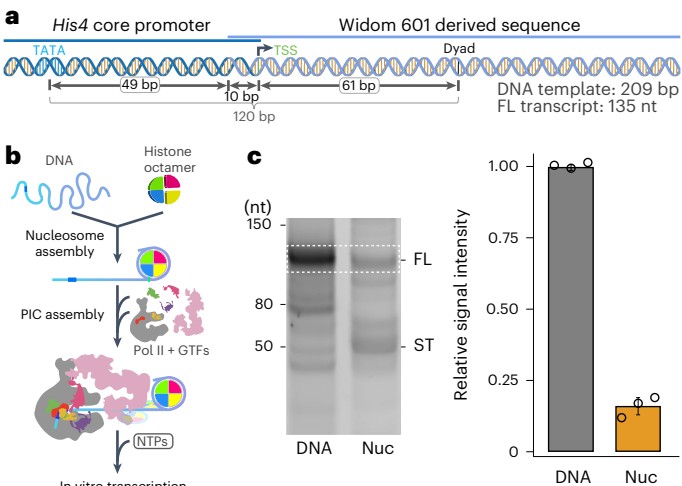

**Fig. 1 | Reconstituted, functional PIC–nucleosome complex. a**, Schematic of template DNA. The distances between the TATA box, TSS and nucleosome dyad are indicated. The expected lengths of template and transcript are noted at the bottom right. **b**, Schematic of in vitro transcription assay. For details, see the Methods. **c**, The +1 nucleosome impairs promoter-dependent transcription in vitro. Assays were performed with template DNA without (DNA) or with +1 nucleosome (Nuc). Representative original scan of the urea–PAGE analysis that yielded results presented on the right. RNA transcripts were analyzed by urea-denaturing PAGE and the full-length product was quantified (dashed rectangle). Experiments were performed at least three times. Bars correspond to the mean of three independent experiments; error bars represent the s.d. FL, full-length transcript; ST, shorter transcripts.

transition from complex A to complex B also led to a further detachment of nucleosomal DNA from the histone octamer (Fig. 3b). In the absence of NTPs (complex A), only one turn of terminal nucleosomal DNA (SHL–7 to SHL–6) is detached from the histone octamer surface, whereas in the presence of NTPs (complex B), two turns of nucleosomal DNA are detached (SHL–7 to SHL–5).

## PIC–nucleosome contacts

The rotation of the incoming nucleosome leads to a more intimate association of the PIC and the +1 nucleosome. In the structure of complex A, TFIIH contacts the +1 nucleosome only via its Tfb2-Tfb5 dimerization domain (Fig. 4a and Supplementary Video 1). In complex B, however, TFIIH forms four contact sites with the nucleosome (Fig. 4b and Supplementary Video 1). TFIIH subunits Ssl2 (XPB in human), Tfb2 (p52), Ssl1 (p44) and Tfb4 (p34) all contain charged loop residues that protrude toward the nucleosome in complex B (Fig. 4b). The Ssl2 ATPase domain engages with downstream DNA, whereas the Ssl2 N-terminal extension and clutch domains form a wedge between DNA and the nucleosome (Fig. 4b). This Ssl2 wedge stabilizes the two detached turns of nucleosomal DNA. Tfb2 possesses a lysine-containing loop in its HTH-3 domain (residues 258–270) that approaches the acidic patch of the nucleosome (Fig. 4b). Ssl1 uses a lysine-rich insertion in its RING domain (residues 414–421) to contact nucleosomal DNA around the dyad (Fig. 4b). Finally, Tfb4 uses an extension in its vWA domain (residues 90–105) to reach near the N-terminal region of histone H4 (Fig. 4b). These PIC-nucleosome interactions may counteract further rotation of the +1 nucleosome and impair TFIIH translocase action beyond this state.

## Evidence for a preferred nucleosome orientation

We next asked whether the position of the nucleosome observed in complex B, and its interactions with TFIIH are specific to the DNA sequence used here or whether they may be of more generic nature. Yeast promoters vary with respect to the distance between their TATA

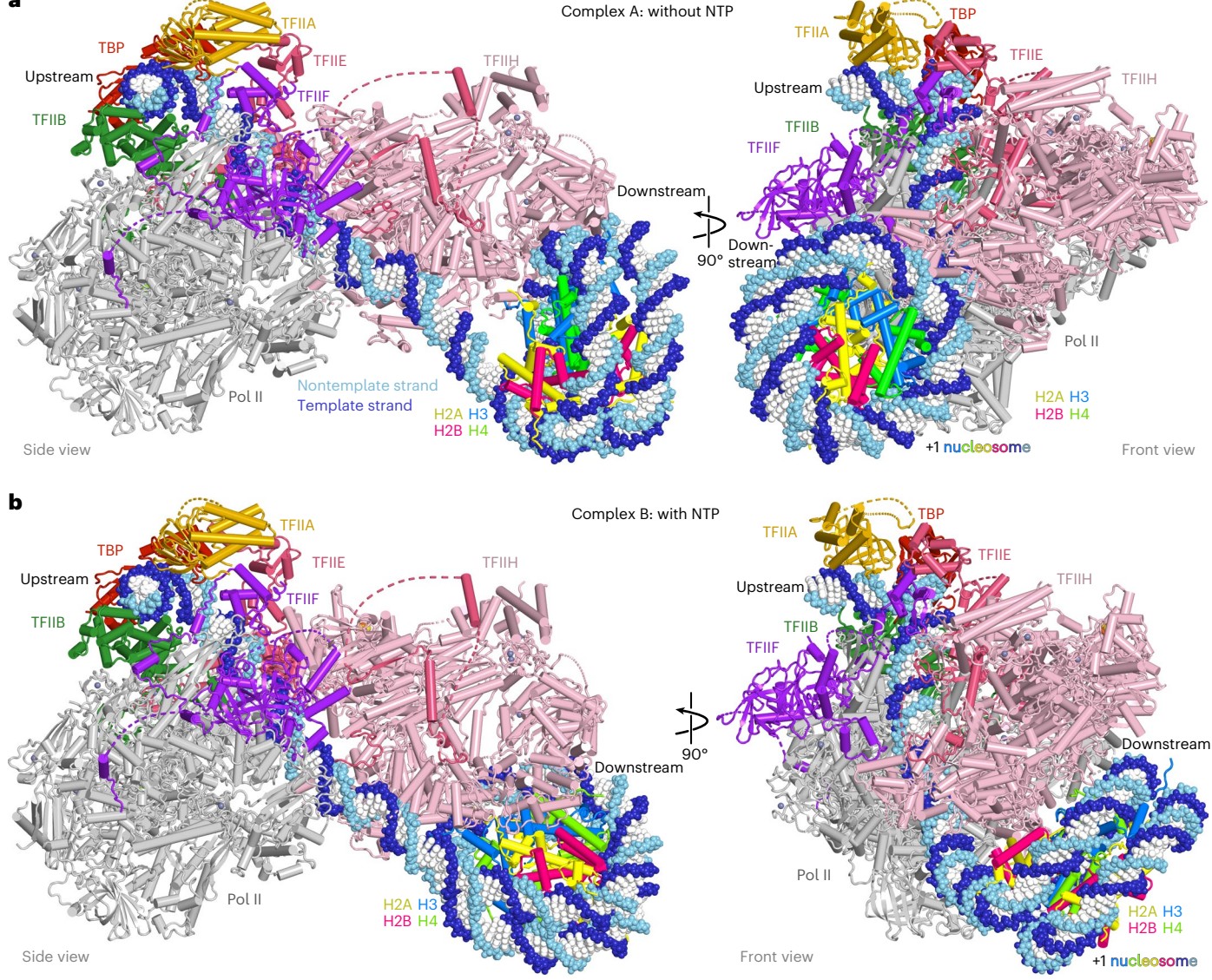

**Fig. 2 | Structures of PIC–nucleosome complexes A and B. a**, Two views of a ribbon model of complex A with template and nontemplate DNA shown as dark and light blue spheres, respectively. **b**, Two views of a ribbon model of complex B with template and nontemplate DNA shown as dark and light blue spheres, respectively.

box and their TSS in a range of roughly 40–120 bp (ref. [26]), and thus the distance between their TATA box and the dyad of the +1 nucleosome can vary in a range of roughly 100–180 bp. To investigate any such distance effects, we prepared an additional PIC–nucleosome complex on an altered DNA template where the nucleosome positioning sequence was shifted downstream by 10 bp (complex C), resulting in a 130 bp distance between the TATA box and the nucleosome dyad, as compared to a 120 bp distance in complexes A and B (Fig. 5a).

We incubated complex C with NTPs and could obtain a cryo-EM structure at an overall resolution of 6.6 Å (Extended Data Fig. 6). Although the resolution prevented us from observing molecular details within the cryo-EM map, the overall orientation of the nucleosome with respect to the PIC in complex C resembled that in complex B (Fig. 5b). This observation indicates that TFIIH uses a common surface to contact the nucleosome in the state of complexes B and C even though the detailed PIC-nucleosome interaction is partially different, suggesting a preferred orientation between TFIIH and the nucleosome may exist when they collide even with different initial PIC–nucleosome distances and on different promoters. We speculate that the preferred orientation of the nucleosome with respect to the PIC observed in complexes

B and C occurs within a common intermediate of the transcription initiation process at yeast promoters.

## Rpb6 N-terminal tail (NTT) in the Pol II active center

In all three PIC–nucleosome structures, we obtained an ordered conformation of the NTT of Pol II subunit Rpb6 (Fig. 6a and Supplementary Video 1) as confirmed by crosslinking mass spectrometry (Extended Data Fig. 7a). Whereas the NTT is mobile in all previous structures of Pol II complexes, Rpb6 residues 12–35 are observed here in the active center cleft of the polymerase (Fig. 6a). The Rpb6 NTT contains several negatively charged residues that interact with positively charged residues in the cleft that are often conserved (Fig. 6a and Extended Data Fig. 7b).

Structural superposition shows that the NTT would clash with DNA and RNA in an initially transcribing complex (ITC)[27] (Fig. 6b), indicating its position is incompatible with transcription. The NTT is dispensable for growth of budding yeast[28] but its mutation causes temperature sensitivity in the fission yeast *S. pombe*[29]. We note that Pol I and Pol III also contain elements that can transiently occupy the active center (Extended Data Fig. 8). These elements are referred to as the expander

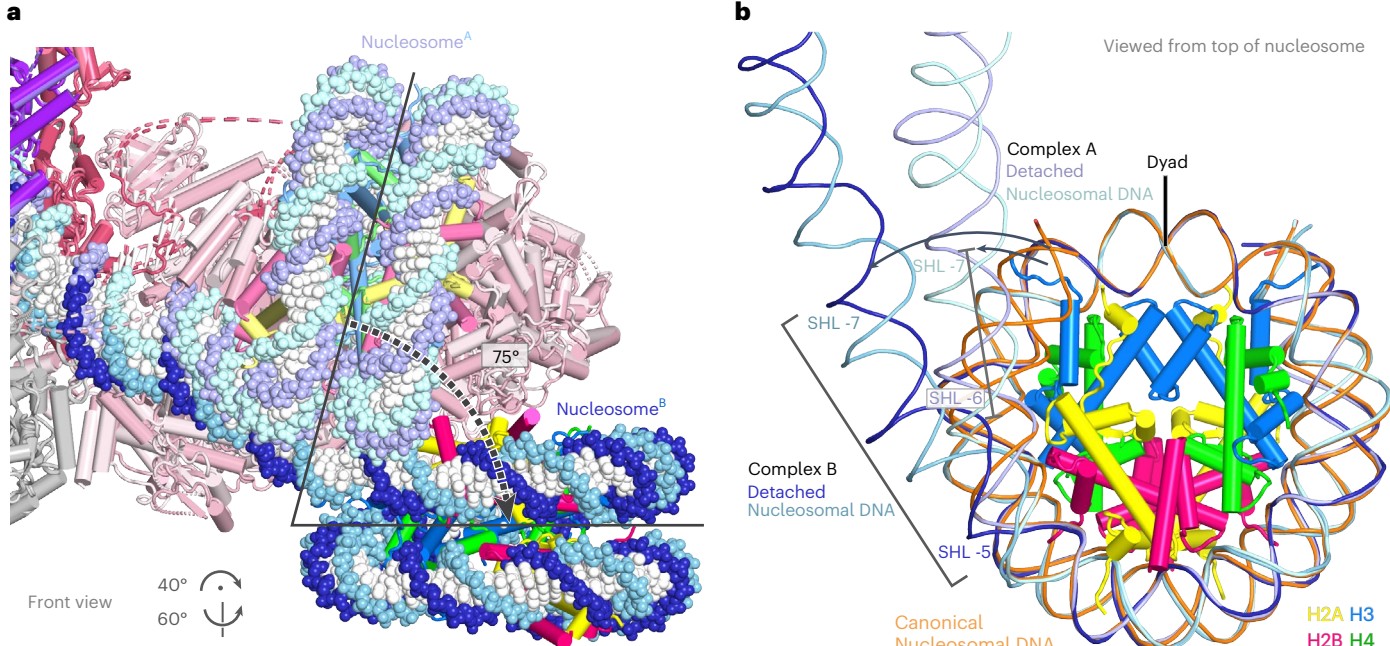

**Fig. 3 | Movement of the +1 nucleosome upon NTP addition. a**, Rotation of the +1 nucleosome as observed by comparison of complex A (light colors, without NTPs) and complex B (full colors, with NTPs). **b**, Detachment of promoter-proximal, terminal nucleosomal DNA from the histone octamer.

Terminal nucleosomal DNA is displaced by roughly 20° and 60° in complexes A and B, respectively, with respect to the canonical nucleosomal DNA path (orange). The nucleosome dyad is indicated in black.

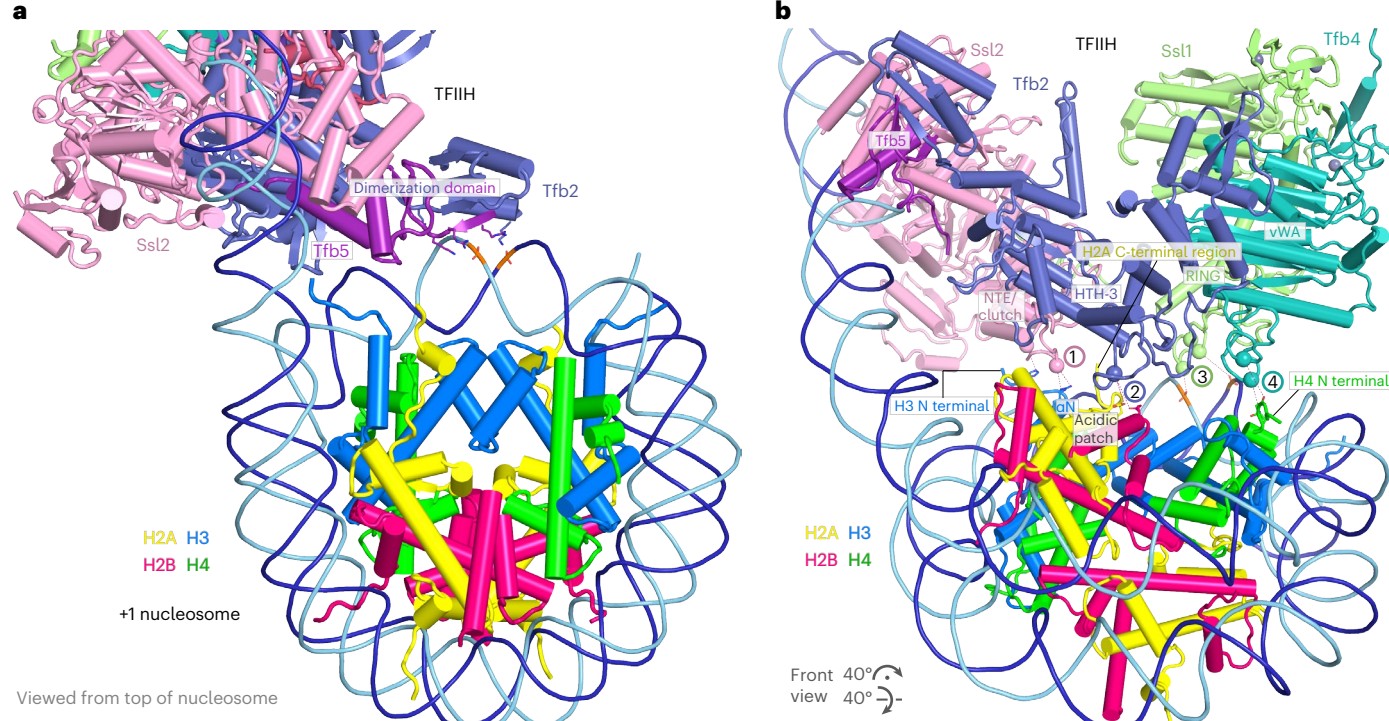

**Fig. 4 | TFIIH–nucleosome contacts. a**, TFIIH–nucleosome interface in complex A. Tfb2 residues K495, K506 and R507, and Tfb5 residues R3, R5 and K6 from the dimerization domain contact DNA around the nucleosome dyad. Except for K495 in Tfb2, these TFIIH residues are conserved in human TFIIH. **b**, TFIIH–nucleosome interface in complex B. Four TFIIH subunits that are implicated in nucleosome contacts are shown in different colors. The view is related to the front view in

Fig. 2a. The first contact may involve Ssl2 residue D103 and H3 residue R52. The second contact may involve Tfb2 residue K262 and the acidic patch on histones H2A and H2B. The third contact may involve Ssl1 residues K414, K417 and K420 that contact DNA around the nucleosome dyad. The fourth contact may involve Tfb4 residue R104 and histone H4 residue D24. Except for R104 in Tfb4 and K414 and K417 in Ssl1, these TFIIH residues are conserved in human TFIIH.

**Table 1 | Cryo-EM data collection, refinement and validation statistics**

| | PIC–nucleosome, complex A (EMD-14927) (PDB 7ZS9) | PIC–nucleosome, complex B (EMD-14928) (PDB 7ZSA) | PIC–nucleosome, complex C (EMD-14929) (PDB 7ZSB) |
|---|---|---|---|
| **Data collection and processing** | | | |
| Magnification | ×81,000 | ×81,000 | ×81,000 |
| Voltage (kV) | 300 | 300 | 300 |
| Electron exposure (e⁻/Å²) | 42 | 41 | 45 |
| Defocus range (µm) | 0.8 to 2.0 | 0.8 to 2.0 | 0.8 to 2.0 |
| Pixel size (Å) | 1.05 | 1.05 | 1.05 |
| Symmetry imposed | C1 | C1 | C1 |
| Initial particle images (no.) | 2,595,857 | 2,329,133 | 1,410,713 |
| Final particle images (no.) | 55,851 | 142,136 | 82,942 |
| Map resolution (Å) | 3.3 | 4.0 | 6.6 |
| FSC threshold | 0.143 | 0.143 | 0.143 |
| Map resolution range (Å) | 2.4–6.5 | 2.9–7.0 | 3.9–8.4 |
| **Refinement** | | | |
| Initial models used (PDB code) | 7O73, 7OHC | 7O73, 7OHC | |
| Model resolution (Å) | 3.2 | 3.7 | |
| FSC threshold | 0.5 | 0.5 | |
| Model resolution range (Å) | 2.5–3.3 | 3.0–3.9 | |
| Map sharpening B factor (Å²) | −60 | −58 | |
| Model composition | | | |
| Nonhydrogen atoms | 86,362 | 86,373 | |
| Protein residues | 9,719 | 9,723 | |
| Nucleotides | 418 | 418 | |
| Ligands | 19 | 19 | |
| B factors (Å²) | | | |
| Protein | 109.51 | 77.55 | |
| Nucleotides | 145.15 | 139.67 | |
| Ligand | 150.14 | 111.24 | |
| R.m.s. deviations | | | |
| Bond lengths (Å) | 0.004 | 0.004 | |
| Bond angles (°) | 0.687 | 0.753 | |
| **Validation** | | | |
| MolProbity score | 1.39 | 1.53 | |
| Clashscore | 5.48 | 6.62 | |
| Poor rotamers (%) | 0.00 | 0.00 | |
| Ramachandran plot | | | |
| Favored (%) | 97.56 | 97.10 | |
| Allowed (%) | 2.44 | 2.90 | |
| Disallowed (%) | 0.00 | 0.00 | |

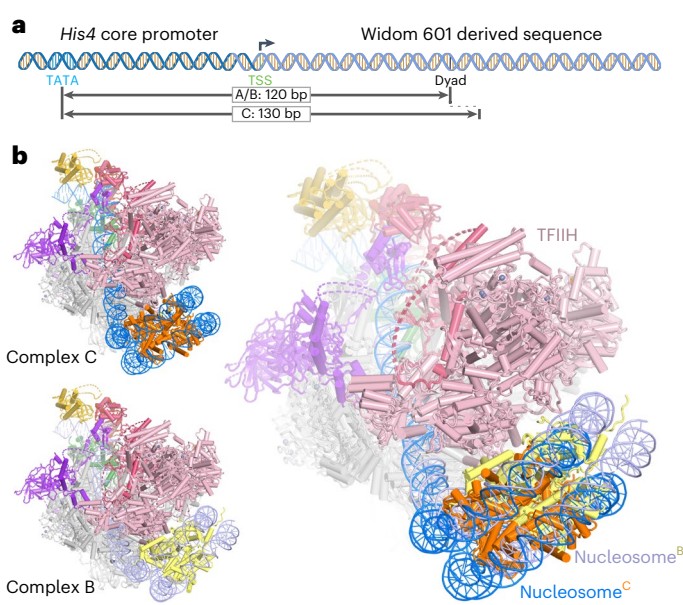

**Fig. 5 | Evidence for a preferred orientation of the +1 nucleosome. a**, Schematic of template DNA for complexes A/B and C. The distances between the TATA box and nucleosome dyad are indicated. **b**, Structure of PIC–nucleosome complex C compared to complex B. The left panel shows the front view of complex C and B. The right panel shows the similar orientation of the +1 nucleosome in each complex aligned on TFIIH.

or the DNA-mimicking loop of A190 in Pol I[30,31], and as the C-terminal tail of RPC7 (C31 in yeast) in Pol III[32]. Further analysis of the function of the NTT awaits the development of a recombinant Pol II system that is currently not available.

### Coactivators may be accommodated

Finally, we asked whether the coactivators Mediator and TFIID may be accommodated in our PIC–nucleosome structures. Superimposition of the yeast core Mediator-PIC structure[4] indeed showed that Mediator could be appended to our structures without clashes (Extended Data Fig. 9a). In the resulting model, the Mediator hook domain approaches the +1 nucleosome up to a distance of roughly 40 Å (Extended Data Fig. 9a). We also superimposed our PIC–nucleosome structures onto two human PIC structures containing TFIID[6,7] since no yeast TFIID-containing PIC structure is available. This showed that TFIID could in principle be accommodated in the PIC–nucleosome complex surface without clashes (Fig. 7 and Extended Data Fig. 9b). The putative TFIID position is consistent with reports that the double bromodomain of TFIID subunit TAF1 and the PHD finger domain of TAF3 can contribute to promoter recognition by binding modified histone tails[33,34] (Extended Data Fig. 9b). Histone modifications might also be involved in regulating the assembly of PIC.

### Discussion

Here, we report structures of Pol II PIC–nucleosome transcription complexes. The structures show that TFIIH directly interacts with the nucleosome in distinct conformations. They also indicate that a preferred orientation of the nucleosome with respect to the PIC is adopted that is characterized by multiple TFIIH–nucleosome contacts. Although the nucleosome may initially be found in different rotational states relative to the PIC, action of the Ssl2 translocase may lead to a preferred nucleosome orientation that allows for multiple TFIIH contacts as observed in complex B and C of this study. As TFIIH apparently has ceased to unravel nucleosomal DNA beyond SHL−5 in complex B, the translocase activity of TFIIH may not be sufficient to enable the PIC to pass through nucleosomal

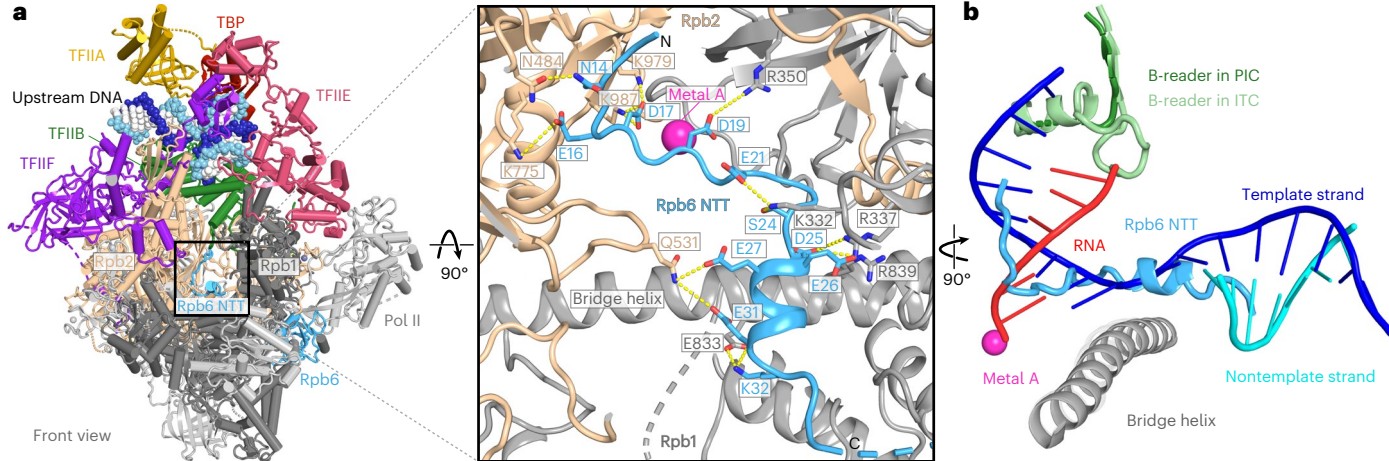

**Fig. 6 | Rbp6 NTT occupies the active center cleft of Pol II. a**, Close-up views of Pol II active center in complex A. Pol II subunits are colored and labeled individually. Rpb6 NTT residues are shown as sticks and labeled as indicated. Electrostatic interactions and hydrogen bonds are shown as yellow dotted lines.

**b**, Superposition of the core PIC containing the Rpb6 NTT with the structure of an ITC (PDB 4BBS)[27] shows that binding of the NTT is incompatible with nucleic acid binding during transcription. The template strand, nontemplate strand and RNA transcript in the ITC structure are indicated.

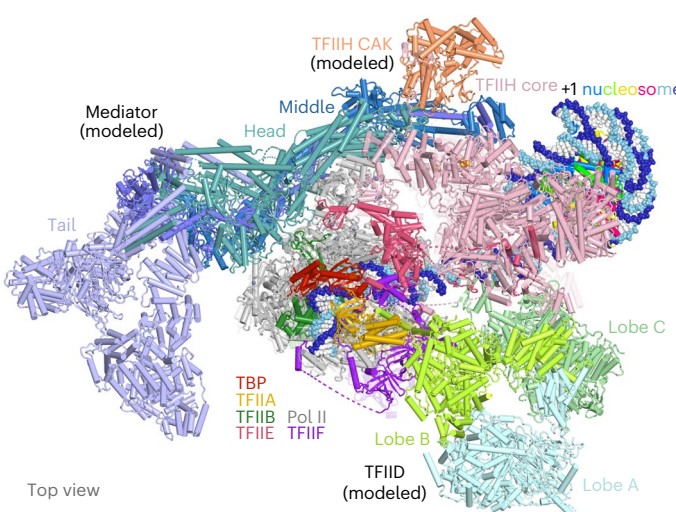

**Fig. 7 | Mediator and TFIID can be accommodated on PIC–nucleosome structure.** Mediator and TFIID were placed onto the complex B PIC–nucleosome structure by superimposing the Mediator- and TFIID-containing human PIC structure (PDB 7ENC)[7] aligned on Pol II.

sites with strong histone-DNA interactions (such as SHL−5 and −1). Passage of PIC through the +1 nucleosome may thus rely on assistance from chromatin remodelers and chromatin modifying complexes.

Together with modeling, the observed structures with the stalled PIC on the nucleosome also provide a possible explanation for why the TSS is often located roughly 60 bp upstream of the dyad of the +1 nucleosome in yeast[15]. Before initiation of RNA chain synthesis, the yeast PIC scans downstream DNA for the TSS[35]. Modeling based on complex B indicates that scanning requires further progression of the PIC into the +1 nucleosome and detachment of three additional turns of DNA (Extended Data Fig. 10). This may be achieved by the ATP-dependent Ssl2 translocase whose activity is required for scanning[36,37]. We speculate that scanning may be impaired at the major histone-DNA interacting sites (for example, SHL−1) just upstream of the nucleosome dyad[23,38], which then may trigger TSS usage and RNA chain initiation as suggested[13]. This model of nucleosome-defined TSS usage may explain why TSSs in yeast are generally located at a

distance of roughly 60 bp from the dyad of the +1 nucleosome, even though the distance from the TSS to the TATA box varies in the range of roughly 40–120 bp (ref. [26]). In our experimental system, the high stability of the nucleosome on the Widom-601 positioning sequence may have prevented scanning and led to a stable intermediate amenable to structure determination.

## Online content

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

## Methods

### Preparation of PIC–nucleosome complex

*S. cerevisiae* 12-subunit Pol II, TBP, TFIIA, TFIIB, TFIIE, TFIIF and TFIIH were purified as described previously[2,4,39]. The DNA scaffolds containing the modified *His4* promoter and Widom-601 sequence were synthesized by Integrated DNA Technologies, amplified and purified as described[40]. *Xenopus laevis* and *S. cerevisiae* histones were prepared and assembled into nucleosomes as described[41,42]. The sequences for the scaffold used in this study is listed with the Widom-601 sequence underlined. The TATA box and TSS are indicated in bold.

*His4* promoter (209 bp):

5′-AGCACGCTGTG**TATATAAT**AGCTATGGAACGTTCGATTCAC CTCCGATGTGTGTTGTACATACATAAAAATATC**A**TAGCTCTTCTGCGCT GTGTTATAGTAATACAATAGTTTACAAAATTTTTTTTCTGAATAATG GTTTTGCCGATTCTACCGTTAATTGATGATCTGGCCTCATGGAATAG TAAGAAGGAATACGTTTCACTTGTT-3′

Scaffold for complex A and B (209 bp):

5′-AGCACGCTGTG**TATATAAT**AGCTATGGAACGTTCGATTCACCTC CGATGTGTGTTGTACATACATAAAAATATC**A**TAGCTCTTCTGCGCTGT GTT<u>GGTCGTAGACAGCTCTAGCACCGCTTAAACGCACGTACGCGCT GTCCCCCGCGTTTTAACCGCCAAGGGGATTACTCCCTAGTCTCCAG GCACGTGTCAGATATATACATCGAT</u>-3′

Scaffold for complex C (219 bp):

5′-AGCACGCTGTG**TATATAAT**AGCTATGGAACGTTCGATTCAC CTCCGATGTGTGTTGTACATACATAAAAATATC**A**TAGCTCTTCTGCGCT GTGTT<u>CCGCTCAATTGGTCGTAGACAGCTCTAGCACCGCTTAAACG CACGTACGCGCTGTCCCCCGCGTTTTAACCGCCAAGGGGATTACTCC CTAGTCTCCAGGCACGTGTCAGATATATACATCGAT</u>-3′

The PIC–nucleosome complex was assembled according to a previously reported protocol[4], with minor modifications. Briefly, scaffold containing a reconstituted nucleosome was incubated with TBP, TFIIA and TFIIB, whereas Pol II and TFIIF were incubated for 10 min at 25 °C. These two preparations were then combined and incubated for another 5 min. TFIIE and preassembled 10-subunit TFIIH were added to the mixture simultaneously and incubated for 5 min. NTPs at a concentration of 400 μM were added and the assembly was incubated for 1 h at 25 °C. The PIC–nucleosome samples were subjected to GraFix ultracentrifugation[43] at 137,600g (32,000 r.p.m. for SW60 rotor) for 16 h at 4 °C in a 15–40% (w/v) sucrose gradient with 0–0.1% glutaraldehyde crosslinker. Subsequently, the gradient solutions were fractionated and quenched with a mixture of 40 mM aspartate and 10 mM lysine for 10 min. Fractions were analyzed by native PAGE. Gels were stained with SYBR Gold (Invitrogen) and Coomassie brilliant blue. Peak fractions containing crosslinked PIC–nucleosome complex were dialyzed for 16 h in dialysis buffer (20 mM HEPES-Na pH 7.5, 50 mM KCl, 2 mM MgCl$_2$, 1 mM TCEP) to remove sucrose. The dialyzed samples were concentrated to 0.2 mg ml$^{-1}$ and used for grid preparation.

### In vitro promoter-dependent transcription assay

Assays were performed as described[9], with minor alterations. DNA scaffolds with or without nucleosome were prepared as described above. All scaffolds contained identical DNA sequences irrespective of the nucleosome component. Assembled scaffolds were stored in a low salt buffer (50 mM KCl, 5 mM K-HEPES pH 7.5, 0.025 mM EDTA). PIC was reconstituted on scaffold DNA essentially as reported[9]. All incubation steps were performed at 25 °C unless indicated otherwise. Per sample, 1.6 pM of TBP, 1.8 pM Pol II, 2.7 pM TFIIE and TFIIH, 9 pM TFIIF, 9 pM TFIIB and 18 pM TFIIA were used. Reactions were prepared in a sample volume of 23.8 μl with final assay conditions of 3 mM HEPES-K pH 7.9, 20 mM Tris-HCl pH 7.9, 60 mM KCl, 8 mM MgCl$_2$, 2% (w/v) PVA, 3% (v/v) glycerol, 0.5 mM DTT, 0.5 mg ml$^{-1}$ BSA and 20 units of RNase inhibitor. Samples were incubated for 45 min and transcription was started by adding 1.2 μl of 10 mM NTP solution and permitted to proceed for 60 min. Reactions were quenched with 100 μl of Stop buffer (300 mM NaCl, 10 mM Tris-HCl pH 7.5, 0.5 mM EDTA) and 14 μl of 10% SDS, followed by

treatment with 4 μg of proteinase K (New England Biolabs) for 30 min at 37 °C. RNA products were isolated from the samples as described[9], applied to urea gels (7 M urea, 1× TBE, 6% acrylamide:bis-acrylamide 19:1) and separated by denaturing gel electrophoresis (urea–PAGE) in 1× TBE buffer for 45 min at 180 V. Gels were stained for 30 min with SYBR Gold (Invitrogen) and RNA was visualized with a Typhoon 9500 FLA imager (GE Healthcare Life Sciences). The densities of the bands on the gels were quantitated with ImageJ.

### Cryo-EM analysis and data processing

Four microliters of PIC–nucleosome samples were applied to glow-discharged UltrAuFoil 2/2 grids (Quantifoil). After incubation on grids for 10 s, samples were blotted for 4 s and vitrified by plunging into liquid ethane via a Vitrobot Mark IV (FEI) operated at 4 °C and 100% humidity. Cryo-EM data were collected on a Titan Krios G2 transmission electron microscope (FEI) operated at 300 keV, equipped with a K3 summit direct detector and a GIF quantum energy filter (Gatan). Automated data acquisition was performed using SerialEM software at a nominal magnification of ×81,000, corresponding to a physical 1.05 Å per pixel. Image stacks of 40 frames were collected in counting mode over 1.5 s at a defocus range from 0.8–2.0 μm. The dose rate was 27 e$^-$/Å$^2$ per second resulting in 1.02 e$^-$/Å$^2$ per frame. Totals of 26,764, 31,286 and 15,515 videos were collected for complexes A, B and C, respectively.

Image stacks were motion-corrected, contrast-transfer function corrected, dose-weighted and auto-picked using Warp[44]. Image processing was performed with RELION v.3.0.5 (ref. [45]). Particles were extracted using a box size of 400$^2$ or 360$^2$ pixels, and normalized. Reference-free 2D classification was performed to remove poorly aligned particles. An ab initio model generated with cryoSPARC[46] was used for subsequent 3D classification. All classes containing PIC–nucleosome density were combined and used for a global 3D refinement. To obtain an improved density map for cPIC, TFIIH and the nucleosome, particles were subjected to focused 3D classification without image alignment. All classes containing good cPIC density were subjected to contrast-transfer function refinement, Bayesian polishing and 3D refinement. Postprocessing of refined models was performed using automated *B* factor determination in RELION and reported resolutions were based on the gold-standard Fourier shell correlation 0.143 criterion. The density of TFIIH was further improved by applying signal subtraction and focused refinement. Local resolution estimates were obtained using the built-in local resolution estimation tool of RELION and previously estimated *B* factors.

### Model building

The structural models were built into the density of the final reconstructions with the best local resolutions for PIC or the TFIIH–nucleosome complex. A nucleosome structure with 145 bp Widom-601 DNA (Protein Data Bank (PDB) 7OHC)[24] and the structure of yeast PIC (PDB 7O73)[9] were placed into the density maps by rigid-body fitting in Chimera[47], followed by manually adjustment and connection of linker DNA. The assignment of Rpb6 NTT was guided by the densities of bulky side chains, the crosslinking mass spectrometry and secondary structure prediction. The models were subjected to alternating real-space refinement and manual adjustment using PHENIX[48] and COOT[49], resulting in very good stereochemistry as assessed by Molprobity[50].

### Crosslinking and mass spectrometry

Samples for mass spectrometry were prepared by sucrose gradient centrifugation as described above for cryo-EM sample preparation without glutaraldehyde. Fractions containing fully assembled complexes were pooled, subjected to chemical crosslinking using zero-length crosslinker EDC (100 mM) and NHS (100 mM) for 1 h at room temperature, and quench with 100 mM ammonium bicarbonate. The samples were adjusted to 8 M urea, 50 mM NH$_4$HCO$_3$, 10 mM DTT followed by an incubation for 30 min at 37 °C. Proteins were

alkylated in the presence of 40 mM iodoacetamide for another 30 min at 37 °C in the dark and the reaction was quenched by 10 mM DTT for 5 min at 37 °C. The reaction volume was adjusted to reach a final concentration of 1 M urea and 50 mM $NH_4HCO_3$. Nucleic acids fragments within the PIC–nucleosome complex were digested for 30 min at 37 °C by the addition of 0.1 M $MgCl_2$ to a final concentration of 1 mM in the reaction and 500 U of universal nuclease (Pierce, catalog no. 88702, 250 U μl$^{-1}$). Trypsin digest was performed overnight at 37 °C with 5 μg of trypsin (Promega, V5111). Peptides were acidified with 4 μl of 100% formic acid, desalted on MicroSpin columns (Harvard Apparatus) following the manufacturer's instructions and vacuum dried. Dried peptides were dissolved in 50 μl of 30% acetonitrile/0.1% TFA and peptide size exclusion (pSEC, Superdex Peptide 3.2/300 column) on an ÄKTA micro system (GE Healthcare) was performed to enrich for crosslinked peptides at a flow rate of 50 μl min$^{-1}$. Fractions of 50 μl were collected. The first 21 fractions enriched in crosslinked peptides were vacuum dried and dissolved in 5% acetonitrile/0.05% TFA (v/v) for analysis by liquid chromatography with tandem mass spectrometry.

Crosslinked peptides derived from pSEC were analyzed as technical duplicates on Q Exactive HF-X hybrid quadrupole-orbitrap mass spectrometer (Thermo Scientific), coupled to a Dionex UltiMate 3000 UHPLC system (Thermo Scientific). The sample was separated on an in-house-packed C18 column (ReproSil-Pur 120 C18-AQ, 1.9 μm pore size, 75 μm inner diameter, 30 cm length, Dr. Maisch GmbH) at a flow rate of 300 nl min$^{-1}$. Sample separation was performed over 60 min using a buffer system consisting of 0.1% (v/v) formic acid (buffer A) and 80% (v/v) acetonitrile, 0.08% (v/v) formic acid (buffer B). The main column was equilibrated with 5% B, followed by sample application and a wash with 5% B. Peptides were eluted by a linear gradient from 15–48% B or 20–50% B. The gradient was followed by a wash step at 95% B and re-equilibration at 5% B. Eluting peptides were analyzed in positive mode using a data-dependent top-30 acquisition methods. MS1 and MS2 resolution were set to 120,000 and 30,000 full width at half maximum, respectively. Precursors selected for MS2 were fragmented using 30% normalized, higher-energy collision induced dissociation fragmentation. Allowed charge states of selected precursors were +3 to +7. Further tandem mass spectrometry parameters were set as follows: isolation width, 1.4 $m/z$; dynamic exclusion, 10 s and maximum injection time (MS1/MS2), 60/200 ms.

For identification of crosslinked peptides, raw files were analyzed by pLink (v.2.3.5), pFind group[51] using EDC as crosslinker and trypsin/P as digestion enzyme with maximal three missed cleavage sites. The search was conducted against a customized protein database containing all proteins within the complex (Supplementary Table 1). Carbamidomethylation of cysteines was set as a fixed modification, oxidation of methionines and acetylation at protein N termini were set as a variable modification. Searches were conducted in combinatorial mode with a precursor mass tolerance of 10 ppm and a fragment ion mass tolerance of 20 ppm. The false discovery rate was set to 0.05 (separate mode). Spectra of both technical duplicates were combined and evaluated manually.

### Reporting summary

Further information on research design is available in the Nature Portfolio Reporting Summary linked to this article.

### Data availability

The electron density reconstructions and final models were deposited with the EM Data Bank (accession codes EMD-14927, 14928 and 14929) and with the PDB (accession codes PDB 7ZS9, 7ZSA and 7ZSB). All mass spectrometry raw files were deposited to the ProteomeXchange Consortium (https://www.proteomexchange.org/) via the PRIDE[52] partner repository with the dataset identifier PRIDE: PXD029840. Source data are provided with this paper.

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

### Acknowledgements

We thank C. Dienemann and U. Steuerwald for advice and for maintaining the EM facility. We thank F. Grabbe for purification of the PIC components and E. Oberbeckmann and L. Xiong for discussions. H.W. was supported by an EMBO Long-Term-Fellowship (grant no. ALTF 650-2017). H.U. was supported by the Deutsche Forschungsgemeinschaft (grant no. SFB860). P.C. was supported by the Deutsche Forschungsgemeinschaft (grant no. SFB860, EXC 2067/1-390729940) and the European Research Council Advanced Investigator grant CHROMATRANS (grant agreement no. 882357).

### Author contributions

H.W. and P.C. conceived the study. H.W. designed and carried out all experiments and data analysis, unless stated otherwise. S.S. performed transcription experiments and provided advice on cryo-EM processing. M.N. and H.U. carried out mass spectrometry analysis. P.C. supervised research. H.W. and P.C. interpreted the data and wrote the manuscript, with input from all authors.

## Funding

## Competing interests

The authors declare no competing interests.

## Additional information

**Extended data** is available for this paper at https://doi.org/10.1038/s41594-022-00865-w.

**Correspondence and requests for materials** should be addressed to Patrick Cramer.

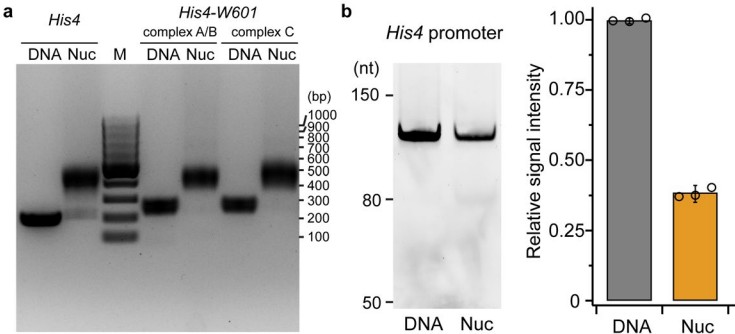

**Extended Data Fig. 1 | Transcription assay using His4 promoter DNA. a**. Reconstitution of nucleosome with *His4* promoter DNAs. Template DNAs (DNA) and reconstituted nucleosomes (Nuc) were analyzed by 1% agarose gel. M: single strand RNA marker with size for each band on the right. **b**. The +1 nucleosome impairs promoter-dependent transcription *in vitro*. Assays were performed with template DNA without (DNA) or with +1 nucleosome (Nuc). RNA transcripts were analyzed by urea-denaturing PAGE and the full-length product was quantified (dashed rectangle). This experiment was repeated 3 times independently with similar results. Bars correspond to the mean of three independent experiments; error bars represent the s.d.

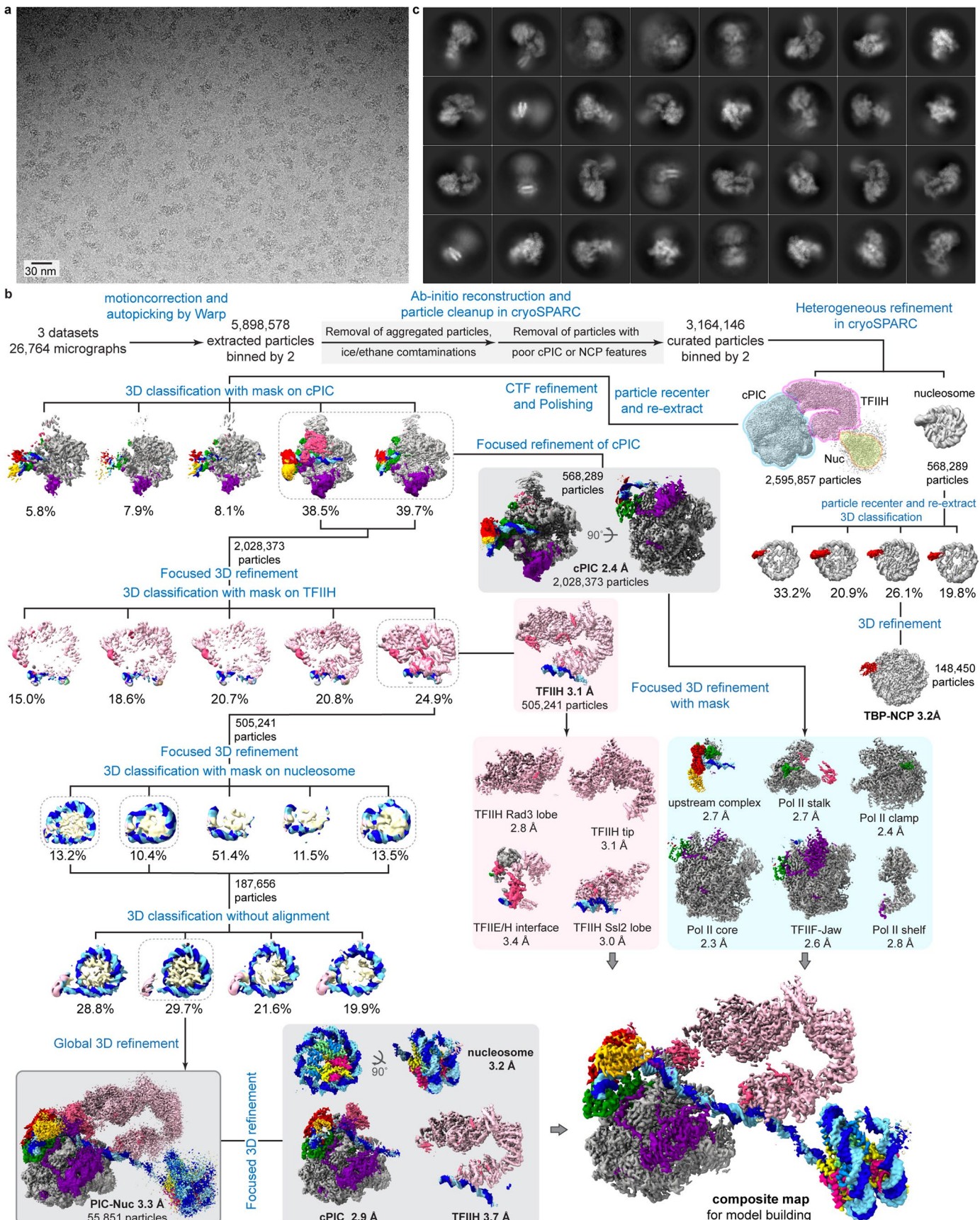

**Extended Data Fig. 2 | See next page for caption.**

**Extended Data Fig. 2 | Cryo-EM structure determination and analysis of PIC-nucleosome complex A. a**. Exemplary cryo-EM micrograph. A scale bar is provided. In total 26,764 micrographs were collected with similar results. **b**. Particle sorting and classification tree. Regions corresponding to Pol II, general transcription factors (GTFs) and template DNA are colored as in Fig. 2a, histones are colored in light yellow. Maps deposited to EMDB are indicated with grey background and outlined in black. The subpopulation of TBP-nucleosome complex is not described in this study since a similar TBP-NCP structure has been published before[24]. **c**. Representative 2D class averages of sorted particles used for final reconstruction.

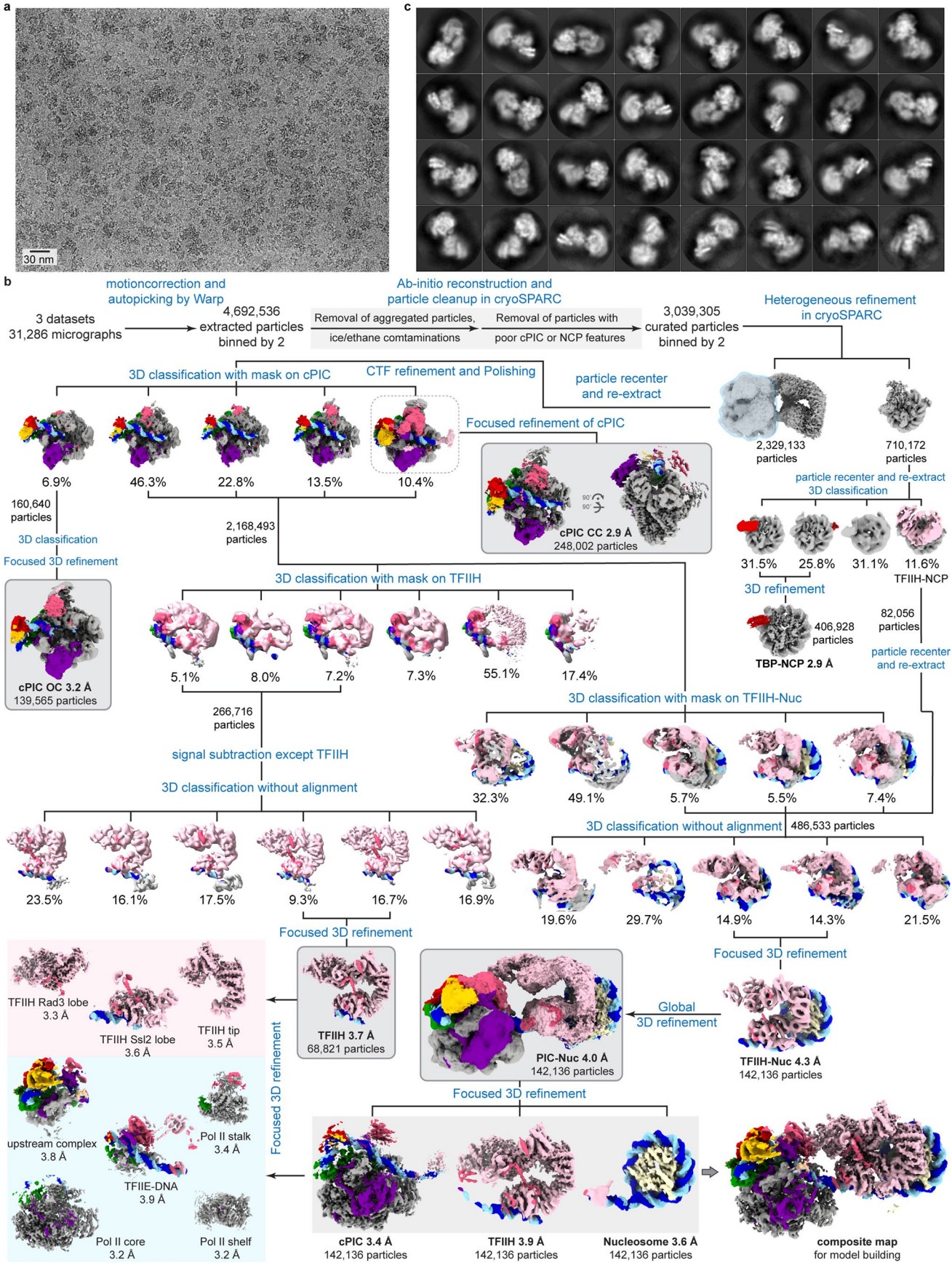

**Extended Data Fig. 3 | See next page for caption.**

**Extended Data Fig. 3 | Cryo-EM structure determination and analysis of PIC-nucleosome complex B. a**. Exemplary cryo-EM micrograph. A scale bar is provided. In total 31,286 micrographs were collected with similar results. **b**. Particle sorting and classification tree. Regions corresponding to Pol II, general transcription factors (GTFs) and template DNA are colored as in Fig. 2b, histones are colored in light yellow. Maps deposited to EMDB are indicated with grey background and outlined in black. The subpopulation of TBP-nucleosome complex is not described in this study since a similar TBP-NCP structure has been published before[24]. **c**. Representative 2D class averages of sorted particles used for final reconstruction.

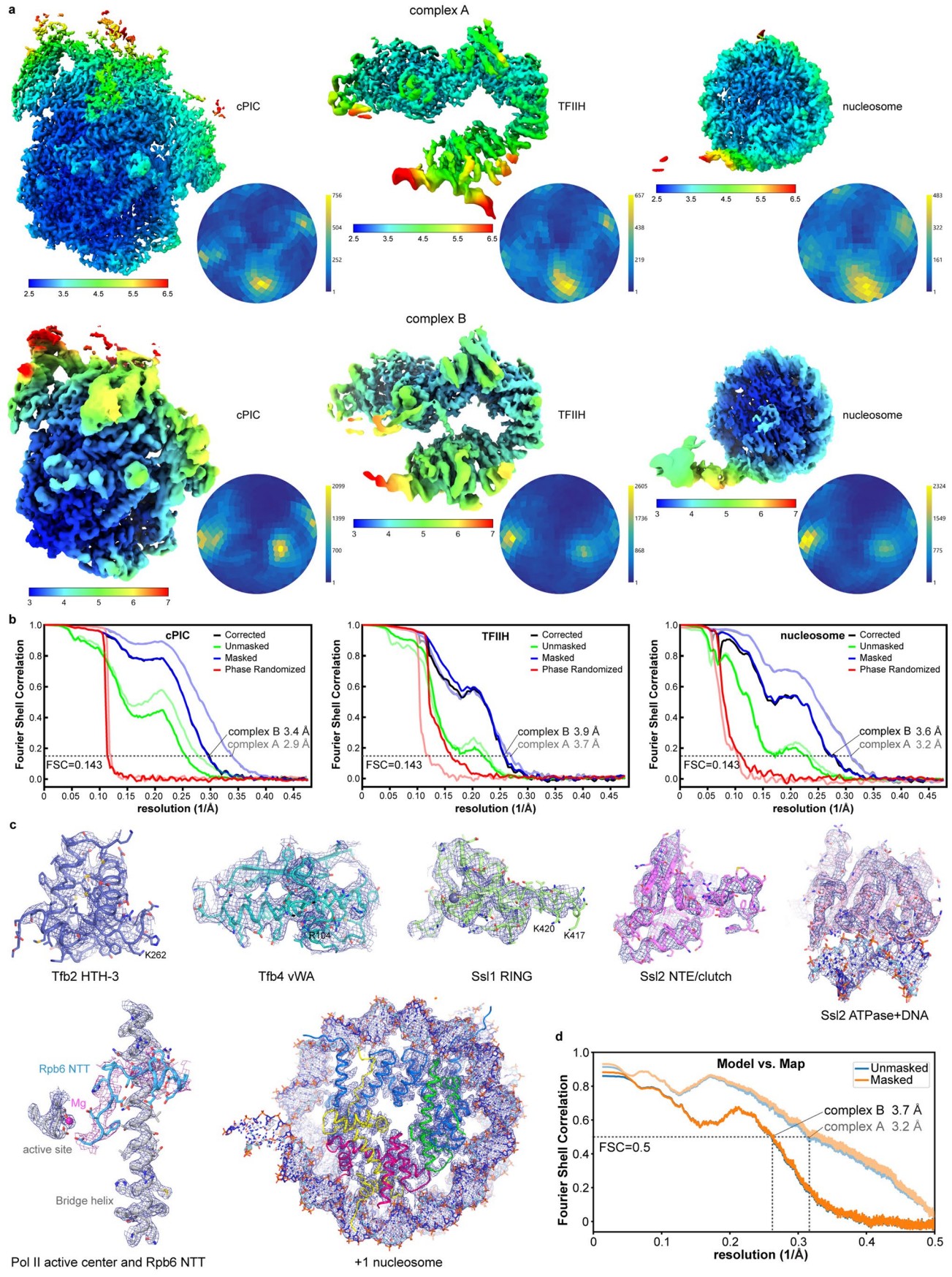

**Extended Data Fig. 4 | See next page for caption.**

**Extended Data Fig. 4 | Quality of cryo-EM reconstructions of complexes A and B. a**. PIC-nucleosome complex A and B reconstructions colored according to local resolution[43] for the core PIC (left), TFIIH (middle) and the +1 nucleosome (right). The color bars from blue to red indicate the local resolution range in Å. Angular distribution diagrams for particles in the final reconstructions are shown on the right. Color shading from blue to yellow correlates with the number of particles at a specific orientation as indicated. **b**. Fourier shell correlation (FSC) between the half maps of the reconstruction. The average resolution is estimated at the FSC 0.143 cut-off criterion (dashed line). **c**. Electron densities (blue mesh) for various parts as indicated. **d**. Model-to-map FSC correlation between the final model and reconstruction. The depicted correlation curves were calculated from the FSC between the derived model and the reconstruction. The resolutions at the FSC 0.5 cut-off criterion (dashed line) are denoted.

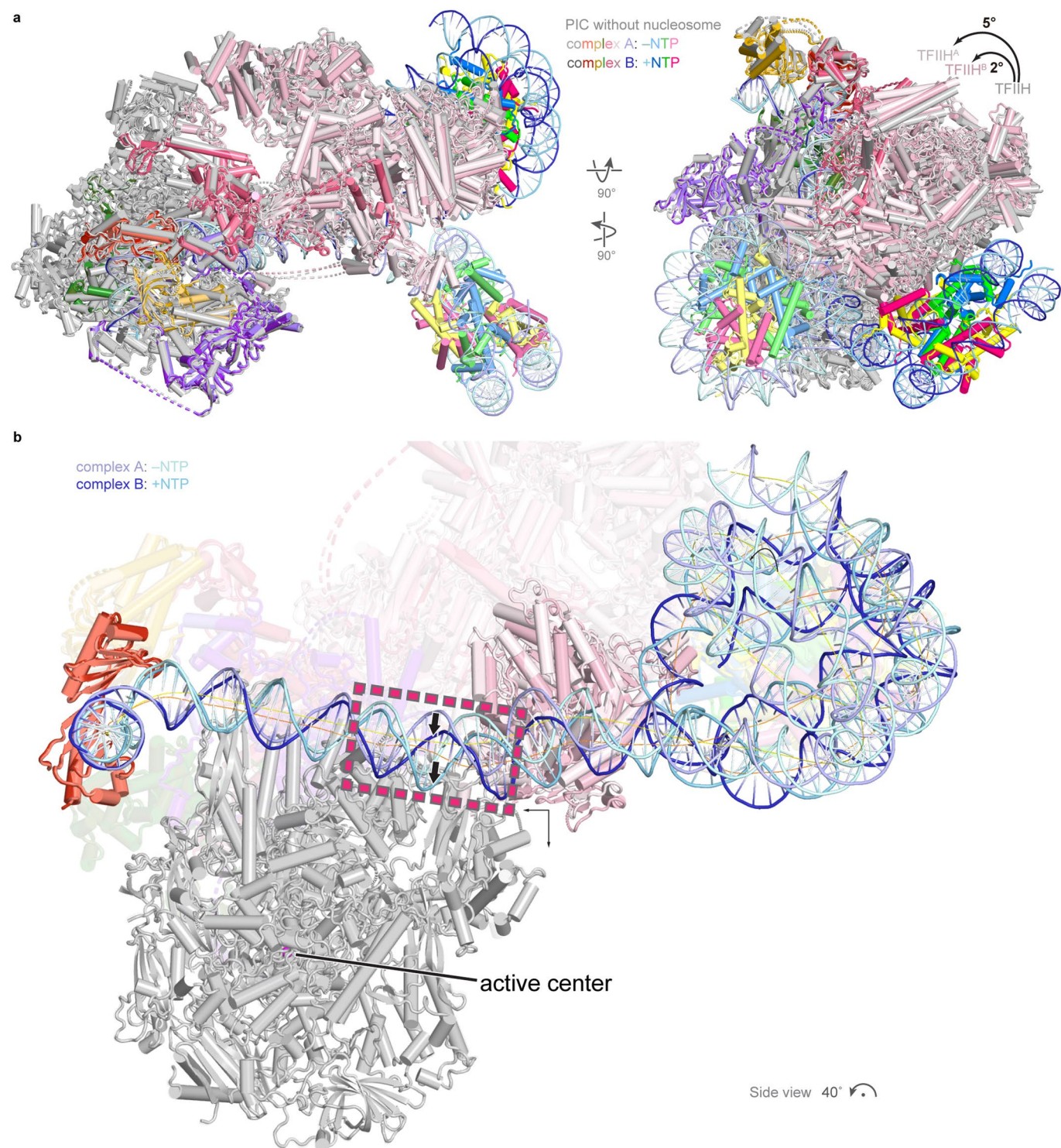

**Extended Data Fig. 5 | Structural changes in the PIC-nucleosome complexes.**
**a**. Superimposition of PIC-nucleosome complex A and B on previous PIC complex (PDB code: 7O73)[9]. **b**. Superimposition of PIC-nucleosome complex A and B aligned on Pol II. The red dashed box encloses the DNA around initially melting region. The black arrows indicate the directions of movement of the DNA towards the active center of Pol II.

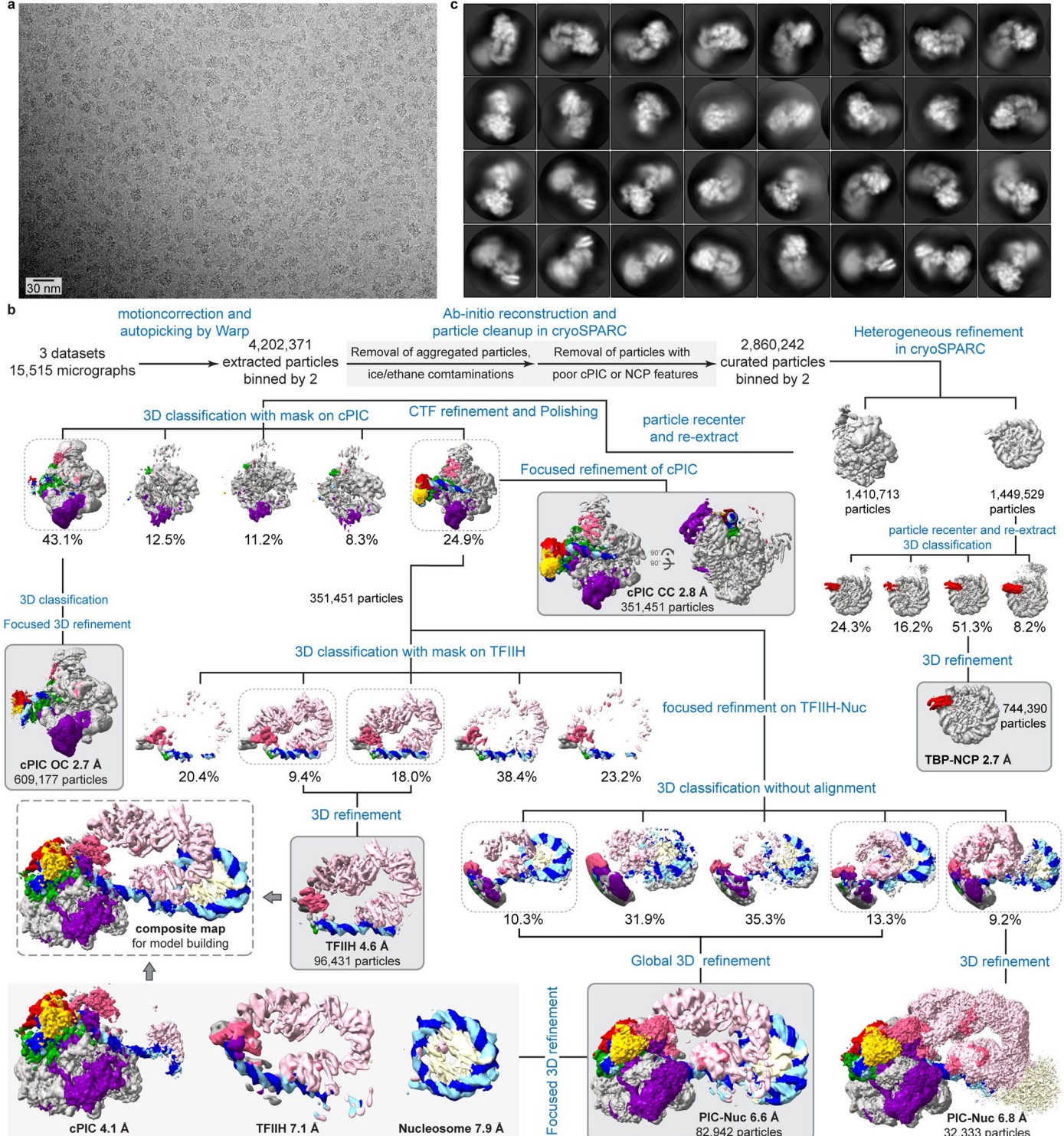

**Extended Data Fig. 6 | Cryo-EM structure determination and analysis of PIC-nucleosome complex C. a**. Exemplary cryo-EM micrograph. A scale bar is provided. In total 15,515 micrographs were collected with similar results. **b**. Particle sorting and classification tree. Regions corresponding to Pol II, general transcription factors (GTFs) and template DNA are colored as in Fig. 2a, histones are colored in light yellow. Maps deposited to EMDB are indicated with grey background and outlined in black. The subpopulation of TBP-nucleosome complex is not described in this study since a similar TBP-NCP structure has been published before[24]. **c**. Representative 2D class averages of sorted particles used for final reconstruction.

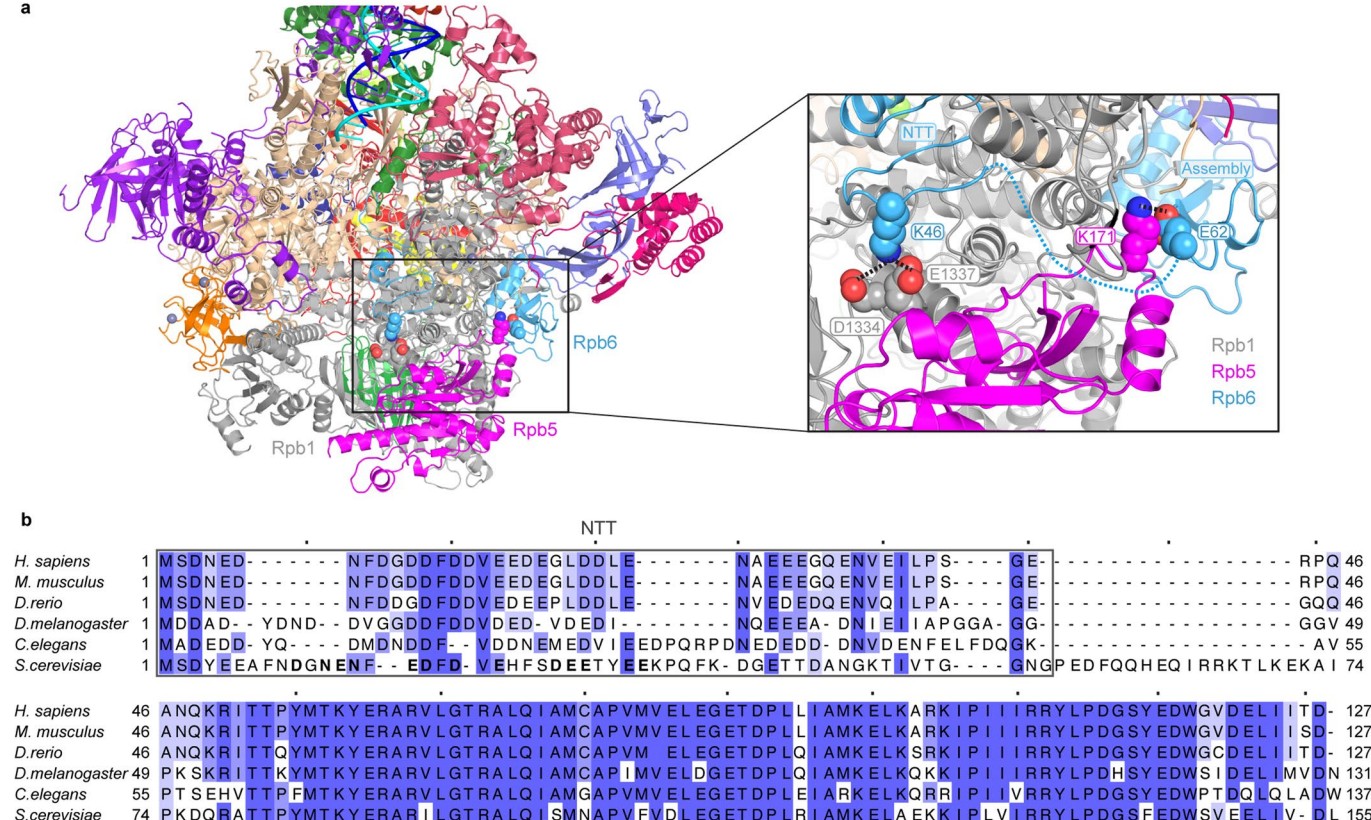

**Extended Data Fig. 7 | Rbp6 NTT occupies the active center cleft of Pol II.**
**a**. Crosslinks between Rpb6 NTT and other Pol II subunits. The close-up view shows three observed EDC crosslinks (black dashed lines). **b**. Sequence alignment of Rpb6 orthologs. The NTT regions are boxed. Acidic residues of Rpb6 NTT that are involved in interaction with the Pol II cleft are indicated in bold.

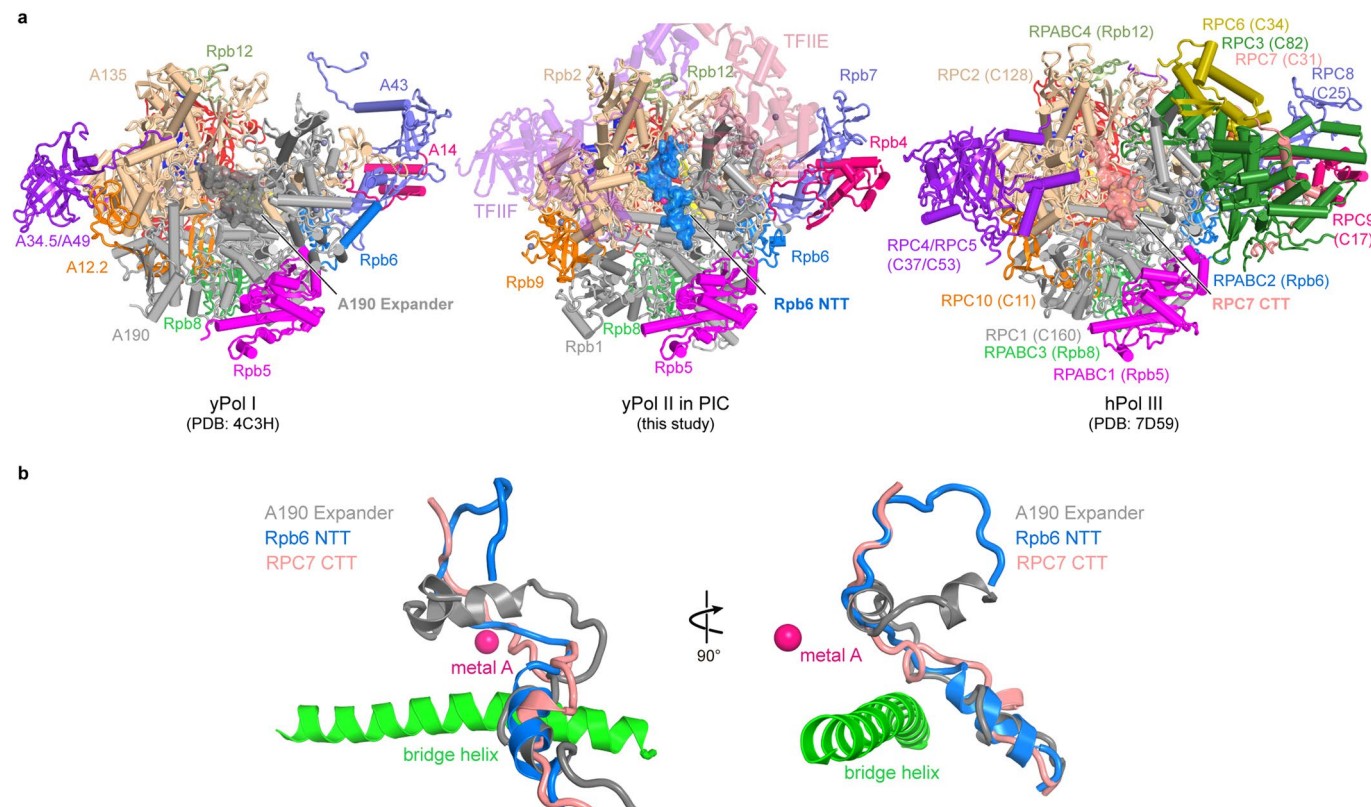

**Extended Data Fig. 8 | Comparison of structural elements occupying the cleft of Pol I, II, and III. a**. Pol I and Pol III contain elements that can occupy the active center cleft at a location corresponding to that observed for the Rpb6 NTT in Pol II (Extended Data Fig. 4). These elements are referred to as the expander or DNA-mimicking loop of A190 in yeast Pol I (PDB code: 4C3H)[31], and as the C-terminal tail of RPC7 (C31 in yeast) in human Pol III (PDB code: 7D59)[32]. **b**. Superposition of cleft elements in Pol I, Pol II and Pol III. The bridge helix and the catalytic magnesium ion in the active site are indicated.

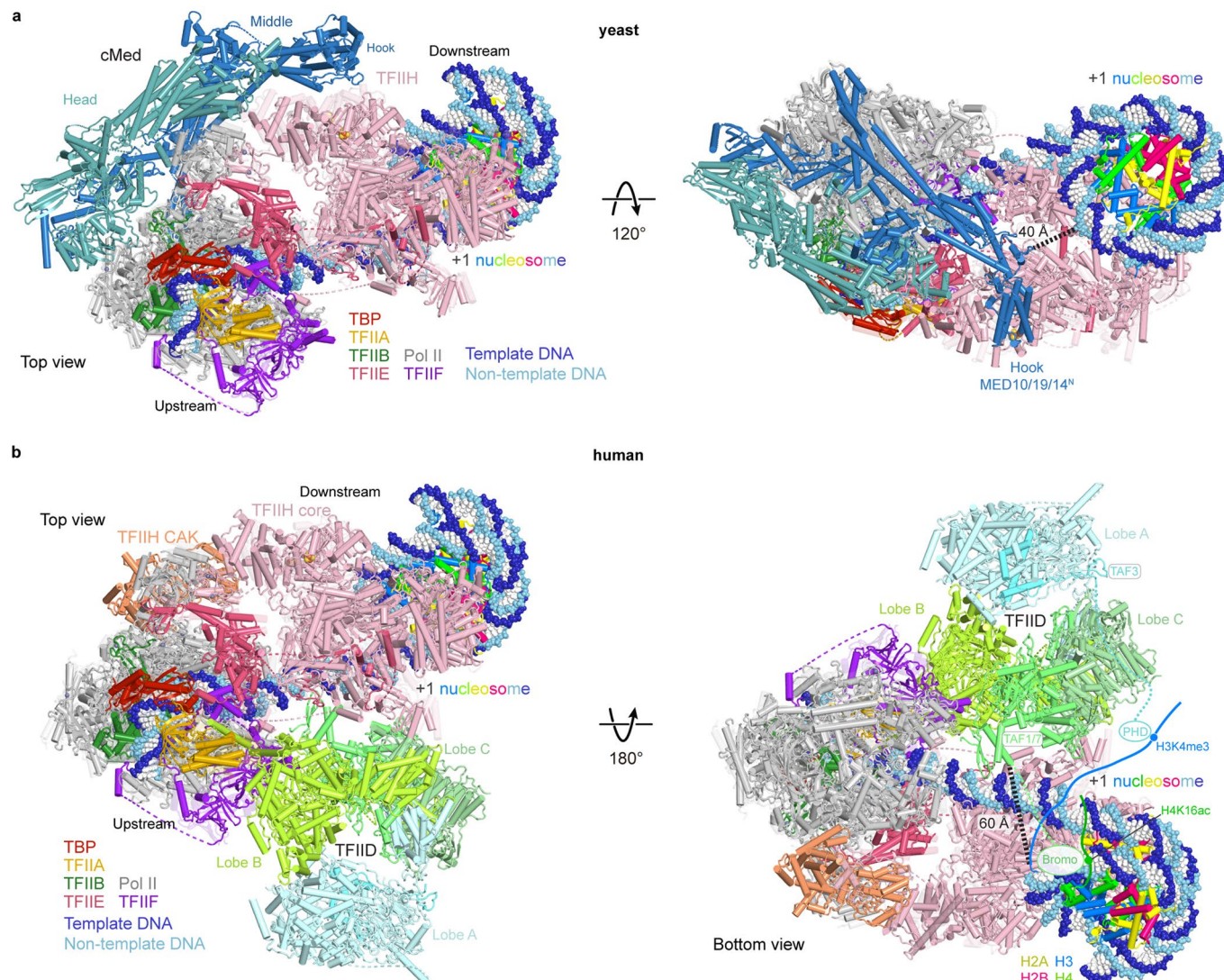

**Extended Data Fig. 9 | Mediator and TFIID are compatible with the PIC-nucleosome structure. a**. Yeast core Mediator (cMed) was placed onto our PIC-nucleosome structure by superimposing the core PIC in the yeast cMed-PIC structure (PDB code: 5OQM)[4]. **b**. TFIID was placed onto the PIC-nucleosome structure by superimposing the core PIC in the human TFIID-containing PIC structure (PDB code: 7EGB)[6] since no yeast TFIID-containing PIC structure is available. The unmodelled double bromodomain of TAF1 and the PHD finger domain of TAF3 are depicted as ovals. The unmodelled histone tails in the structure are depicted as threads with histone modifications symbolized as spheres.

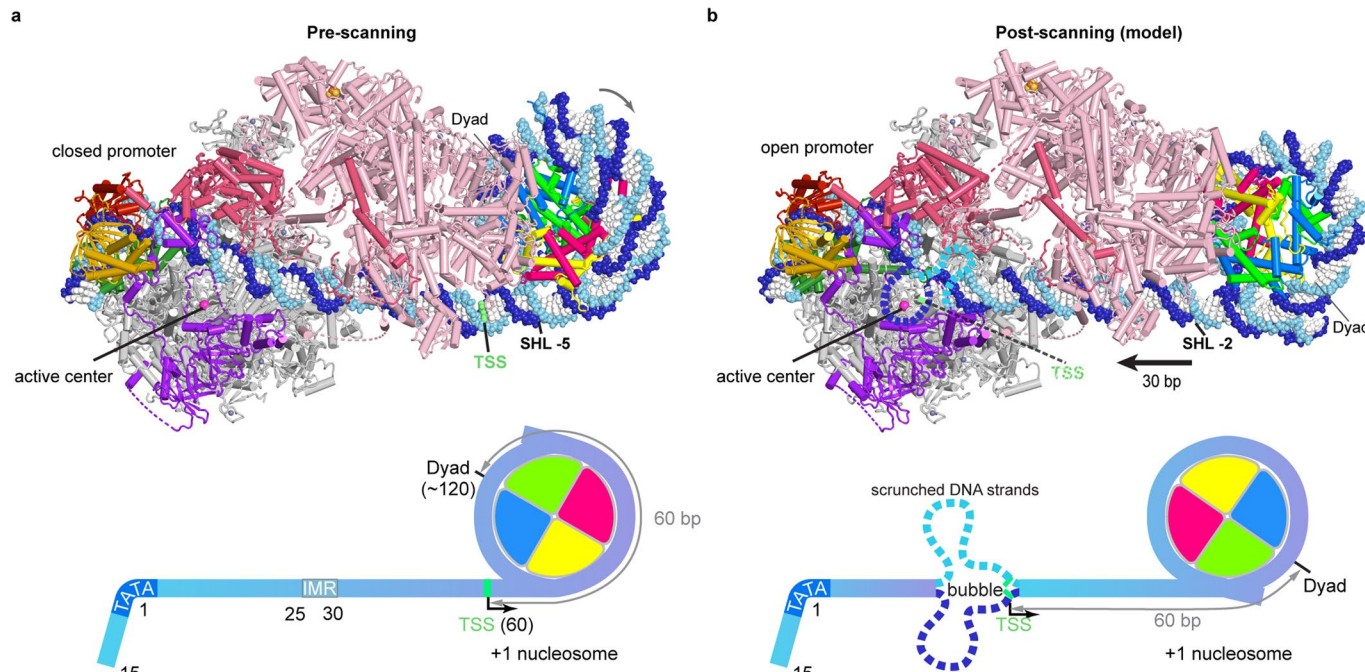

**Extended Data Fig. 10 | Model for TSS scanning. a.** The PIC-nucleosome complex structure presented here corresponds to the state before scanning starts (pre-scanning state). The Pol II active center and the TSS are indicated. The lower panel shows a schematic representation. Distances between the TATA box and selected DNA elements are denoted. **b.** Model of the PIC-nucleosome complex in the post-scanning state. The modelled structure was generated using structures of the yeast PIC with open DNA (PDB code: 7O75)[9] and of Pol II with a partially unraveled nucleosome (PDB code: 6A5T)[23]. The partially unraveled nucleosome with 5 turns of nucleosomal DNA detached from the histone octamer position was aligned with the PIC-nucleosome structure based on DNA and manually adjusted to avoid clashes between the nucleosome and TFIIH.

# Reporting Summary

## Statistics

For all statistical analyses, confirm that the following items are present in the figure legend, table legend, main text, or Methods section.

| n/a | Confirmed | |
|---|---|---|
| ☐ | ☒ | The exact sample size (*n*) for each experimental group/condition, given as a discrete number and unit of measurement |
| ☒ | ☐ | A statement on whether measurements were taken from distinct samples or whether the same sample was measured repeatedly |
| ☒ | ☐ | The statistical test(s) used AND whether they are one- or two-sided *Only common tests should be described solely by name; describe more complex techniques in the Methods section.* |
| ☒ | ☐ | A description of all covariates tested |
| ☒ | ☐ | A description of any assumptions or corrections, such as tests of normality and adjustment for multiple comparisons |
| ☐ | ☒ | A full description of the statistical parameters including central tendency (e.g. means) or other basic estimates (e.g. regression coefficient) AND variation (e.g. standard deviation) or associated estimates of uncertainty (e.g. confidence intervals) |
| ☒ | ☐ | For null hypothesis testing, the test statistic (e.g. *F*, *t*, *r*) with confidence intervals, effect sizes, degrees of freedom and *P* value noted *Give P values as exact values whenever suitable.* |
| ☒ | ☐ | For Bayesian analysis, information on the choice of priors and Markov chain Monte Carlo settings |
| ☒ | ☐ | For hierarchical and complex designs, identification of the appropriate level for tests and full reporting of outcomes |
| ☒ | ☐ | Estimates of effect sizes (e.g. Cohen's *d*, Pearson's *r*), indicating how they were calculated |

*Our web collection on statistics for biologists contains articles on many of the points above.*

## Software and code

Policy information about availability of computer code

| Data collection | Serial EM 3.8 beta 8 |
|---|---|
| Data analysis | RELION 3.0.5, UCSF ChimeraX 1.2.5, Coot 0.9, Warp 1.0.7, PHENIX 1.19, cryoSPARC 2.14.2, ImageJ 1.53 |

For manuscripts utilizing custom algorithms or software that are central to the research but not yet described in published literature, software must be made available to editors and reviewers. We strongly encourage code deposition in a community repository (e.g. GitHub). See the Nature Portfolio guidelines for submitting code & software for further information.

## Data

Policy information about availability of data

All manuscripts must include a data availability statement. This statement should provide the following information, where applicable:

- Accession codes, unique identifiers, or web links for publicly available datasets
- A description of any restrictions on data availability
- For clinical datasets or third party data, please ensure that the statement adheres to our policy

The electron density reconstructions and final models were deposited with the EM Data Base (accession code: EMD-14927, 14928 and 14929) and with the Protein Data Bank (accession code 7ZS9, 7ZSA and 7ZSB). All mass spectrometry raw files were deposited to the ProteomeXchange Consortium (https://www.proteomexchange.org/) via the PRIDE52 partner repository with the dataset identifier PRIDE: PXD029840. Source data are provided with this paper.

# Field-specific reporting

Please select the one below that is the best fit for your research. If you are not sure, read the appropriate sections before making your selection.

☒ Life sciences  ☐ Behavioural & social sciences  ☐ Ecological, evolutionary & environmental sciences

For a reference copy of the document with all sections, see nature.com/documents/nr-reporting-summary-flat.pdf

# Life sciences study design

All studies must disclose on these points even when the disclosure is negative.

| | |
|---|---|
| Sample size | Cryo-EM micrographs were collected until about 4-5 millions particles were autopicked by Warp for each dataset |
| Data exclusions | No data were excluded from the analyses. |
| Replication | All attempts at replication were successful. Cryo-EM single particle analysis inherently relies on averaging over a large number of independent observations. The transcription assays were carried out with at least 3 independent replicates. |
| Randomization | Samples were not allocated to groups. |
| Blinding | Blinding is not a common practice for cryo-EM studies. |

# Behavioural & social sciences study design

All studies must disclose on these points even when the disclosure is negative.

| | |
|---|---|
| Study description | Briefly describe the study type including whether data are quantitative, qualitative, or mixed-methods (e.g. qualitative cross-sectional, quantitative experimental, mixed-methods case study). |
| Research sample | State the research sample (e.g. Harvard university undergraduates, villagers in rural India) and provide relevant demographic information (e.g. age, sex) and indicate whether the sample is representative. Provide a rationale for the study sample chosen. For studies involving existing datasets, please describe the dataset and source. |
| Sampling strategy | Describe the sampling procedure (e.g. random, snowball, stratified, convenience). Describe the statistical methods that were used to predetermine sample size OR if no sample-size calculation was performed, describe how sample sizes were chosen and provide a rationale for why these sample sizes are sufficient. For qualitative data, please indicate whether data saturation was considered, and what criteria were used to decide that no further sampling was needed. |
| Data collection | Provide details about the data collection procedure, including the instruments or devices used to record the data (e.g. pen and paper, computer, eye tracker, video or audio equipment) whether anyone was present besides the participant(s) and the researcher, and whether the researcher was blind to experimental condition and/or the study hypothesis during data collection. |
| Timing | Indicate the start and stop dates of data collection. If there is a gap between collection periods, state the dates for each sample cohort. |
| Data exclusions | If no data were excluded from the analyses, state so OR if data were excluded, provide the exact number of exclusions and the rationale behind them, indicating whether exclusion criteria were pre-established. |
| Non-participation | State how many participants dropped out/declined participation and the reason(s) given OR provide response rate OR state that no participants dropped out/declined participation. |
| Randomization | If participants were not allocated into experimental groups, state so OR describe how participants were allocated to groups, and if allocation was not random, describe how covariates were controlled. |

# Ecological, evolutionary & environmental sciences study design

All studies must disclose on these points even when the disclosure is negative.

| | |
|---|---|
| Study description | Briefly describe the study. For quantitative data include treatment factors and interactions, design structure (e.g. factorial, nested, hierarchical), nature and number of experimental units and replicates. |
| Research sample | Describe the research sample (e.g. a group of tagged Passer domesticus, all Stenocereus thurberi within Organ Pipe Cactus National Monument), and provide a rationale for the sample choice. When relevant, describe the organism taxa, source, sex, age range and any manipulations. State what population the sample is meant to represent when applicable. For studies involving existing datasets, describe the data and its source. |

| Sampling strategy | *Note the sampling procedure. Describe the statistical methods that were used to predetermine sample size OR if no sample-size calculation was performed, describe how sample sizes were chosen and provide a rationale for why these sample sizes are sufficient.* |
|---|---|
| Data collection | *Describe the data collection procedure, including who recorded the data and how.* |
| Timing and spatial scale | *Indicate the start and stop dates of data collection, noting the frequency and periodicity of sampling and providing a rationale for these choices. If there is a gap between collection periods, state the dates for each sample cohort. Specify the spatial scale from which the data are taken* |
| Data exclusions | *If no data were excluded from the analyses, state so OR if data were excluded, describe the exclusions and the rationale behind them, indicating whether exclusion criteria were pre-established.* |
| Reproducibility | *Describe the measures taken to verify the reproducibility of experimental findings. For each experiment, note whether any attempts to repeat the experiment failed OR state that all attempts to repeat the experiment were successful.* |
| Randomization | *Describe how samples/organisms/participants were allocated into groups. If allocation was not random, describe how covariates were controlled. If this is not relevant to your study, explain why.* |
| Blinding | *Describe the extent of blinding used during data acquisition and analysis. If blinding was not possible, describe why OR explain why blinding was not relevant to your study.* |

Did the study involve field work? ☐ Yes ☐ No

## Field work, collection and transport

| Field conditions | *Describe the study conditions for field work, providing relevant parameters (e.g. temperature, rainfall).* |
|---|---|
| Location | *State the location of the sampling or experiment, providing relevant parameters (e.g. latitude and longitude, elevation, water depth).* |
| Access & import/export | *Describe the efforts you have made to access habitats and to collect and import/export your samples in a responsible manner and in compliance with local, national and international laws, noting any permits that were obtained (give the name of the issuing authority, the date of issue, and any identifying information).* |
| Disturbance | *Describe any disturbance caused by the study and how it was minimized.* |

# Reporting for specific materials, systems and methods

We require information from authors about some types of materials, experimental systems and methods used in many studies. Here, indicate whether each material, system or method listed is relevant to your study. If you are not sure if a list item applies to your research, read the appropriate section before selecting a response.

## Materials & experimental systems

| n/a | Involved in the study |
|---|---|
| ☒ | ☐ Antibodies |
| ☒ | ☐ Eukaryotic cell lines |
| ☒ | ☐ Palaeontology and archaeology |
| ☒ | ☐ Animals and other organisms |
| ☒ | ☐ Human research participants |
| ☒ | ☐ Clinical data |
| ☒ | ☐ Dual use research of concern |

## Methods

| n/a | Involved in the study |
|---|---|
| ☒ | ☐ ChIP-seq |
| ☒ | ☐ Flow cytometry |
| ☒ | ☐ MRI-based neuroimaging |

## Antibodies

| Antibodies used | *Describe all antibodies used in the study; as applicable, provide supplier name, catalog number, clone name, and lot number.* |
|---|---|
| Validation | *Describe the validation of each primary antibody for the species and application, noting any validation statements on the manufacturer's website, relevant citations, antibody profiles in online databases, or data provided in the manuscript.* |

## Palaeontology and Archaeology

| Specimen provenance | *Provide provenance information for specimens and describe permits that were obtained for the work (including the name of the issuing authority, the date of issue, and any identifying information). Permits should encompass collection and, where applicable, export.* |
|---|---|
| Specimen deposition | *Indicate where the specimens have been deposited to permit free access by other researchers.* |

| Dating methods | *If new dates are provided, describe how they were obtained (e.g. collection, storage, sample pretreatment and measurement), where they were obtained (i.e. lab name), the calibration program and the protocol for quality assurance OR state that no new dates are provided.* |

☐ Tick this box to confirm that the raw and calibrated dates are available in the paper or in Supplementary Information.

| Ethics oversight | *Identify the organization(s) that approved or provided guidance on the study protocol, OR state that no ethical approval or guidance was required and explain why not.* |

Note that full information on the approval of the study protocol must also be provided in the manuscript.

# Animals and other organisms

Policy information about studies involving animals; ARRIVE guidelines recommended for reporting animal research

| Laboratory animals | *For laboratory animals, report species, strain, sex and age OR state that the study did not involve laboratory animals.* |

| Wild animals | *Provide details on animals observed in or captured in the field; report species, sex and age where possible. Describe how animals were caught and transported and what happened to captive animals after the study (if killed, explain why and describe method; if released, say where and when) OR state that the study did not involve wild animals.* |

| Field-collected samples | *For laboratory work with field-collected samples, describe all relevant parameters such as housing, maintenance, temperature, photoperiod and end-of-experiment protocol OR state that the study did not involve samples collected from the field.* |

| Ethics oversight | *Identify the organization(s) that approved or provided guidance on the study protocol, OR state that no ethical approval or guidance was required and explain why not.* |

Note that full information on the approval of the study protocol must also be provided in the manuscript.

# Human research participants

Policy information about studies involving human research participants

| Population characteristics | *Describe the covariate-relevant population characteristics of the human research participants (e.g. age, gender, genotypic information, past and current diagnosis and treatment categories). If you filled out the behavioural & social sciences study design questions and have nothing to add here, write "See above."* |

| Recruitment | *Describe how participants were recruited. Outline any potential self-selection bias or other biases that may be present and how these are likely to impact results.* |

| Ethics oversight | *Identify the organization(s) that approved the study protocol.* |

Note that full information on the approval of the study protocol must also be provided in the manuscript.

# Clinical data

Policy information about clinical studies

All manuscripts should comply with the ICMJE guidelines for publication of clinical research and a completed CONSORT checklist must be included with all submissions.

| Clinical trial registration | *Provide the trial registration number from ClinicalTrials.gov or an equivalent agency.* |

| Study protocol | *Note where the full trial protocol can be accessed OR if not available, explain why.* |

| Data collection | *Describe the settings and locales of data collection, noting the time periods of recruitment and data collection.* |

| Outcomes | *Describe how you pre-defined primary and secondary outcome measures and how you assessed these measures.* |

# Dual use research of concern

Policy information about dual use research of concern

## Hazards

Could the accidental, deliberate or reckless misuse of agents or technologies generated in the work, or the application of information presented in the manuscript, pose a threat to:

| No | Yes | |
|----|-----|--|
| ☐ | ☐ | Public health |
| ☐ | ☐ | National security |
| ☐ | ☐ | Crops and/or livestock |
| ☐ | ☐ | Ecosystems |
| ☐ | ☐ | Any other significant area |

## Experiments of concern

Does the work involve any of these experiments of concern:

| No | Yes | |
|----|-----|--|
| ☐ | ☐ | Demonstrate how to render a vaccine ineffective |
| ☐ | ☐ | Confer resistance to therapeutically useful antibiotics or antiviral agents |
| ☐ | ☐ | Enhance the virulence of a pathogen or render a nonpathogen virulent |
| ☐ | ☐ | Increase transmissibility of a pathogen |
| ☐ | ☐ | Alter the host range of a pathogen |
| ☐ | ☐ | Enable evasion of diagnostic/detection modalities |
| ☐ | ☐ | Enable the weaponization of a biological agent or toxin |
| ☐ | ☐ | Any other potentially harmful combination of experiments and agents |

# ChIP-seq

## Data deposition

☐ Confirm that both raw and final processed data have been deposited in a public database such as GEO.

☐ Confirm that you have deposited or provided access to graph files (e.g. BED files) for the called peaks.

| | |
|---|---|
| **Data access links**<br>*May remain private before publication.* | *For "Initial submission" or "Revised version" documents, provide reviewer access links.  For your "Final submission" document, provide a link to the deposited data.* |
| **Files in database submission** | *Provide a list of all files available in the database submission.* |
| **Genome browser session**<br>(e.g. UCSC) | *Provide a link to an anonymized genome browser session for "Initial submission" and "Revised version" documents only, to enable peer review.  Write "no longer applicable" for "Final submission" documents.* |

## Methodology

| | |
|---|---|
| **Replicates** | *Describe the experimental replicates, specifying number, type and replicate agreement.* |
| **Sequencing depth** | *Describe the sequencing depth for each experiment, providing the total number of reads, uniquely mapped reads, length of reads and whether they were paired- or single-end.* |
| **Antibodies** | *Describe the antibodies used for the ChIP-seq experiments; as applicable, provide supplier name, catalog number, clone name, and lot number.* |
| **Peak calling parameters** | *Specify the command line program and parameters used for read mapping and peak calling, including the ChIP, control and index files used.* |
| **Data quality** | *Describe the methods used to ensure data quality in full detail, including how many peaks are at FDR 5% and above 5-fold enrichment.* |
| **Software** | *Describe the software used to collect and analyze the ChIP-seq data. For custom code that has been deposited into a community repository, provide accession details.* |

# Flow Cytometry

## Plots

Confirm that:

☐ The axis labels state the marker and fluorochrome used (e.g. CD4-FITC).

☐ The axis scales are clearly visible. Include numbers along axes only for bottom left plot of group (a 'group' is an analysis of identical markers).

☐ All plots are contour plots with outliers or pseudocolor plots.

☐ A numerical value for number of cells or percentage (with statistics) is provided.

## Methodology

| | |
|---|---|
| Sample preparation | *Describe the sample preparation, detailing the biological source of the cells and any tissue processing steps used.* |
| Instrument | *Identify the instrument used for data collection, specifying make and model number.* |
| Software | *Describe the software used to collect and analyze the flow cytometry data. For custom code that has been deposited into a community repository, provide accession details.* |
| Cell population abundance | *Describe the abundance of the relevant cell populations within post-sort fractions, providing details on the purity of the samples and how it was determined.* |
| Gating strategy | *Describe the gating strategy used for all relevant experiments, specifying the preliminary FSC/SSC gates of the starting cell population, indicating where boundaries between "positive" and "negative" staining cell populations are defined.* |

☐ Tick this box to confirm that a figure exemplifying the gating strategy is provided in the Supplementary Information.

# Magnetic resonance imaging

## Experimental design

| | |
|---|---|
| Design type | *Indicate task or resting state; event-related or block design.* |
| Design specifications | *Specify the number of blocks, trials or experimental units per session and/or subject, and specify the length of each trial or block (if trials are blocked) and interval between trials.* |
| Behavioral performance measures | *State number and/or type of variables recorded (e.g. correct button press, response time) and what statistics were used to establish that the subjects were performing the task as expected (e.g. mean, range, and/or standard deviation across subjects).* |

## Acquisition

| | |
|---|---|
| Imaging type(s) | *Specify: functional, structural, diffusion, perfusion.* |
| Field strength | *Specify in Tesla* |
| Sequence & imaging parameters | *Specify the pulse sequence type (gradient echo, spin echo, etc.), imaging type (EPI, spiral, etc.), field of view, matrix size, slice thickness, orientation and TE/TR/flip angle.* |
| Area of acquisition | *State whether a whole brain scan was used OR define the area of acquisition, describing how the region was determined.* |

Diffusion MRI     ☐ Used          ☐ Not used

## Preprocessing

| | |
|---|---|
| Preprocessing software | *Provide detail on software version and revision number and on specific parameters (model/functions, brain extraction, segmentation, smoothing kernel size, etc.).* |
| Normalization | *If data were normalized/standardized, describe the approach(es): specify linear or non-linear and define image types used for transformation OR indicate that data were not normalized and explain rationale for lack of normalization.* |
| Normalization template | *Describe the template used for normalization/transformation, specifying subject space or group standardized space (e.g. original Talairach, MNI305, ICBM152) OR indicate that the data were not normalized.* |
| Noise and artifact removal | *Describe your procedure(s) for artifact and structured noise removal, specifying motion parameters, tissue signals and physiological signals (heart rate, respiration).* |

| Volume censoring | *Define your software and/or method and criteria for volume censoring, and state the extent of such censoring.* |

## Statistical modeling & inference

| Model type and settings | *Specify type (mass univariate, multivariate, RSA, predictive, etc.) and describe essential details of the model at the first and second levels (e.g. fixed, random or mixed effects; drift or auto-correlation).* |

| Effect(s) tested | *Define precise effect in terms of the task or stimulus conditions instead of psychological concepts and indicate whether ANOVA or factorial designs were used.* |

Specify type of analysis: ☐ Whole brain ☐ ROI-based ☐ Both

Statistic type for inference
(See Eklund et al. 2016)

*Specify voxel-wise or cluster-wise and report all relevant parameters for cluster-wise methods.*

| Correction | *Describe the type of correction and how it is obtained for multiple comparisons (e.g. FWE, FDR, permutation or Monte Carlo).* |

## Models & analysis

| n/a | Involved in the study |
| --- | --- |
| ☐ | ☐ Functional and/or effective connectivity |
| ☐ | ☐ Graph analysis |
| ☐ | ☐ Multivariate modeling or predictive analysis |

Functional and/or effective connectivity

*Report the measures of dependence used and the model details (e.g. Pearson correlation, partial correlation, mutual information).*

Graph analysis

*Report the dependent variable and connectivity measure, specifying weighted graph or binarized graph, subject- or group-level, and the global and/or node summaries used (e.g. clustering coefficient, efficiency, etc.).*

Multivariate modeling and predictive analysis

*Specify independent variables, features extraction and dimension reduction, model, training and evaluation metrics.*

