## [Peer Review File · Nature Structural & Molecular Biology]

Peer Review Information

Manuscript Title: Structures of transcription preinitiation complex engaged with the +1 nucleosome

Corresponding author name(s): Professor Patrick Cramer

Editorial Notes:

Transferred manuscripts	This manuscript has been previously reviewed at another journal that is not operating a transparent peer review scheme. This document only contains reviewer comments, rebuttal and decision letters for versions considered at Nature XX .
Transferred manuscripts (no peer review at Nature XX)	This manuscript has been previously reviewed at another journal that is not operating a transparent peer review scheme. The manuscript was considered suitable for publication without further review at Nature XX .
Redactions – transferred manuscripts (mention of previous referee reports from elsewhere)	This manuscript has been previously reviewed at another journal. This document only contains reviewer comments, rebuttal and decision letters for versions considered at Nature XX . Mentions of prior referee reports have been redacted
Redactions – transferred manuscripts (mention of the other journal)	This manuscript has been previously reviewed at another journal. This document only contains reviewer comments, rebuttal and decision letters for versions considered at Nature XX . Mentions of the other journal have been redacted.
Redactions – unpublished data	Parts of this Peer Review File have been redacted as indicated to maintain the confidentiality of unpublished data.
Redactions – confidential patient information	Parts of this Peer Review File have been redacted as indicated to maintain patient confidentiality.
Redactions – published data	Parts of this Peer Review File have been redacted as indicated to remove third-party material.
Redactions – reviewer opt-out	Parts of this Peer Review File have been redacted as indicated as we could not obtain permission to publish the reports of reviewer no. XX .
Reviewer comments in marked-up manuscript	In their review of the [first/second/third/...] version of this manuscript, reviewer no. XX added their comments to the manuscript file. These comments, excluding minor textual revisions, have been copied into this Peer Review File.

Reviewer Comments & Decisions:

EA, duplicate for each version as needed, and then delete this instruction.

Decision Letter, initial version:

Message: Our ref: NSMB-A46269-T

19th Jul 2022

Dear Dr. Cramer,

Thank you for submitting your revised manuscript entitled "Structures of transcription preinitiation complex engaged with the +1 nucleosome" (NSMB-A46269-T). It has now been seen by the original referees and their comments are below. The reviewers raise some pertinent concerns but overall they find that the paper has improved in revision, and therefore we'll be happy in principle to publish it in Nature Structural & Molecular Biology, pending revisions to fully satisfy the referees' final requests and to comply with our editorial and formatting guidelines. No additional experiments are required but we think that some of the claims/interpretations need to be toned down. We may need to consult with the reviewers one final time to make sure that they are satisfied with the revised manuscript.

We are now performing detailed checks on your paper and will send you a checklist detailing our editorial and formatting requirements in about two weeks. Please do not upload the final materials and make any revisions until you receive this additional information from us. In the meantime, however, we encourage you to start addressing the reviewers' comments.

Thank you again for your interest in Nature Structural & Molecular Biology. Please do not hesitate to contact me if you have any questions.

Sincerely,

Tiago

Tiago Faial, PhD
Consulting Editor
Nature Structural & Molecular Biology

Reviewer #1 (Remarks to the Author):

The revised paper from Wang et al. contains two additional structures that give more

context to the structure (now Complex B) from the earlier version. This is helpful in that it shows the +1 nucleosome is not fixed relative to the PIC, validating the concern I raised in my first review. I have no technical issues with the work here, which lives up to the high standards of the Cramer lab. However, I still think the authors' interpretations go beyond what the data really show.

They still claim they've shown "intimate interactions" (abstract) between TFIID and the +1 nucleosome that "stabilize the PIC" (line 199). However, complexes A and B have completely different orientations and contact points (really just proximities, given the low resolutions). Complexes B and C only form after TFIID is already incorporated into the PIC and beginning to translocate. What effect would any PIC stabilization have at this late point? If anything, the inhibition of transcription suggests collision with the nucleosome destabilizes the PIC to ITC conversion by impeding progression to an open complex. The +1 nucleosome might have a positive effect via TFIID bromodomain interactions (which is the mechanism suggested in reference 11), but in the context shown here it clearly has a negative effect.

Multiple reviewers suggested that mutagenesis of the proposed TFIID contact points could help test the authors' model, but the authors say those experiments aren't feasible. Without functional validation, the contact points could just be where TFIID and the nucleosome collide as the translocase begins to pull in the DNA. In their reply, the authors seem to be moving in this direction ("we do not think that TFIID specifically recognizes the nucleosome, rather, it serves as a docking site that restricts nucleosome orientation"), but this doesn't come across in the paper, which seems to suggest a stable interaction.

The observation that Complex C, with ten extra base pairs, shows a similar TFIID-nucleosome orientation as Complex B could be significant, or it might just be because the non-native Widom 601 positioning sequence is only easily unraveled up to this point (as the authors suggest in the last sentences of the discussion). With only the current data, I don't think you can claim this is "a stable intermediate state that gets populated during the initiation phase". It could be one of many transient intermediate rotational positions of the nucleosome that we don't see, artifactually stabilized by the 601 sequence and crosslinking.

Line 201. I don't know of any data suggesting that TFIID activity is affected by histone modifications. Is there any basis for this speculation? Would this be a direct or indirect effect?

Is the slight TFIID rotation in Complex B similar to that reported by Murakami in the PIC species leading to ITC (Yang et al., Mol Cell 2022)? If so, that would be worth noting.

In sum, I think the work here is technically impressive, but the authors are stuck on a model of a positive TFIID-nucleosome interaction. There's no functional data to support this positive model, and the *in vitro* transcription inhibition even argues against it. The structures are interesting and worth publishing if the authors were more careful in their interpretation.

Reviewer #2 (Remarks to the Author):

The authors addressed in a satisfactory way all my comments and suggestions. They provide additional structures that confirm their hypothesis and strengthen the manuscript. In my opinion this work is ready to be published in NSMB.

Reviewer #3 (Remarks to the Author):

In this paper, Wang et al. present the structure of the PIC with an adjacent +1 nucleosome. They determined three structures that are potential intermediates in the collision of the PIC with the +1 nucleosome. The results are interesting and thought provoking, but the authors' many claims need further studies to determine. To dissect this, I will give comment on several of the major claims in the discussion of their paper first. Then go through some the major issues I see in the results.

We report here the first structures of Pol II PIC-nucleosome transcription complexes.

- True, but this is only a partial PIC. Though the reason for this is provided by the author was that TFIID was too flexible in their attempts generate a complex. This could be interesting data to show in the paper.

The structures show that TFIID interacts with the nucleosome and that a preferred orientation of the nucleosome with respect to the PIC may exist that is characterized by multiple TFIID-nucleosome contacts.

- First the structure does appear to show multiple contact between TFIID and the nucleosome however it would seem that each structure appears to show different contacts and as such a "preferred orientation" would not appear to be true.

- Additionally, the contacts they do observe I question the importance of, they are not observed at high enough resolution to show any meaningful interaction (no specific interacting residue pairs can be seen).

These contacts suggest that the +1 nucleosome can help to stabilize the PIC via TFIID interactions and may explain how PIC formation can be influenced by the +1 nucleosome.

- Is the nucleosome-PIC interaction stable? The three states show different interactions. It would seem that the template used for the assembly of these complexes are a bigger factor for how the nucleosome will interact with the PIC.

The location of TFIID in the vicinity of histone tails might allow the regulation of TFIID activity by histone modifications.

- Is this a known phenomenon? Please give reference. I think this is the first time in the paper this is mentioned.

Although the nucleosome may initially be found at different rotational states with respect to the PIC, it is likely that action of the Ssl2 translocase leads to a preferred nucleosome orientation that allows for multiple TFIID contacts as observed here.

- Would it not be possible that if a differently spaced template was used that TFIID could also stabilize this interaction? Complex C used a 10 bp extended template that would give the nucleosome the same rotational state as complex B. Even still it appears that TFIID interact with the nucleosome in a different manner.

The observation that TFIID stops unravelling nucleosomal DNA at SHL -5 indicates that the translocase activity of TFIID is not enough to enable PIC to pass through the strong histone-DNA interacting sites (such as SHL -5 and -1) within nucleosome.

- Is this true in the both the A/B and C template containing structures? The comparison of Complex B and C needs to be expanded in the text and figures. It appears in Figure 5 that

the TFIID-nucleosome interactions are different.

Passage of PIC through the +1 nucleosome may require the help from chromatin remodelers and chromatin modifying complexes.

- While not a major point are the authors stating that a chromatin modifier directly helps the passage of the PIC through the +1 or that having a modified nucleosome would help?
- Secondly, would not the acetylation (if this is the type of modification the authors are thinking about) of the +1 typically precede the assembly of the PIC? This would lead to the bigger question as to whether a modified nucleosome would stall the PIC, given that acetylated nucleosome is thought to be slightly less stable than unmodified ones.

Together with modeling, the structures also provide an explanation for why the transcription start site (TSS) is often located ~60 bp upstream of the dyad of the +1 nucleosome in yeast.

- This is the first sentence of the second and last paragraph of the discussion. This claim is meant to be supported by the rest of this paragraph. I do not see how this claim is at all supported by data.

To initiate RNA chain synthesis, the yeast PIC scans downstream DNA for the TSS.

- While this may seem obvious, the how has never been determined. No one has yet shown the nature of this scanning and how the PIC can start at the TATA box and make its way to the TSS.
- What the authors capture is the beginning of this scanning, and to explain "why the transcription start site (TSS) is often located ~60 bp upstream of the dyad of the +1 nucleosome in yeast" they would need to show the TSS in the active site of the polymerase.

Modeling based on complex B indicates that scanning requires further progression of the PIC into the +1 nucleosome and detachment of three additional turns of DNA (Extended Data Fig. 10). This may be achieved by the ATP-dependent Ssl2 translocase that is required for scanning.

- Clearly this is not enough, or else you would see the promoter open.

We speculate that scanning may be impaired at the major nucleosomal barrier just upstream of the nucleosome dyad, which then may trigger TSS usage and RNA chain initiation as suggested.

- What do the authors mean by "major nucleosomal barrier"? The nucleosome itself or a specific DNA-octamer interaction (e.g. SHL -1)?

This model of nucleosome defined TSS usage may explain why TSSs in yeast are generally located at a distance of ~60 bp from the dyad of the +1 nucleosome, whereas the distance from the TSS to the TATA box varies in the range of ~40-120 bp.

- Yes, the model laid out can explain "why TSSs in yeast are generally located at a distance of ~60 bp from the dyad of the +1 nucleosome," but it does not explain why "the TSS to the TATA box varies in the range of ~40-120 bp." Like I stated before the "No one has yet shown the nature of this scanning and how the PIC can start at the TATA box and make its way to the TSS"

In our experimental system, the high stability of the nucleosome on the Widom-601 positioning sequence may have prevented scanning and led to a stable intermediate amenable to structure determination.

- Possibly, and could also have prevented the authors from explaining "why the transcription start site (TSS) is often located ~60 bp upstream of the dyad of the +1 nucleosome in yeast"

Future studies should focus on the roles of Mediator and TFIID in nucleosome-involved transcription initiation.

While the discussion section needs to be rewritten to better represent the actual finding of this paper. I also have a major concerns with the in vitro transcription assay results and the representation and description of the Complex C state in the results.

Concerns:

In Supplemental Figure 1 (the one that corresponds to Figure 1c) how is the length of the products determined? The gel does not appear to have a ladder. Additionally, FL product is presumably between 80-150bp and would mean that the TSS is locate within the Widom601 sequence. If this is the case, why then is the TSS the same for DNA and Nuc? Is it 60bp form the nucleosome dyad and when there is not nucleosome 60bp after the TATA? And these just happen to be the same place? What would occur with the complex C template? Previously results like from (Nagai et al. PNAS 2017: doi: 10.1073/pnas.1620312114) indicate that given the conditions of this experiment that there should be some product starting at ~30bp form the TATA.

In Supplemental Figure 1 (the one that corresponds to Extended Data Figure 1b) I have the same product size concern as the previous figure. The other concern I have is that the boxed lanes show a noticeable reduction in the band intensity for the right (Nuc) then any of the other bands. Presumably the other lanes are not just DNA and so which leads mean to wonder which are Nuc and why do they look no way near the intensity of the selected band. The in especially concerning given the supposed deviation of the band intensity is very low based on the circle values shown in Extended Data Figure 1b.

Comment:

A more detailed comparison is need for Complex B and C. Both in the text and in the Figure 4. In the text it is stated "overall orientation of the nucleosome with respect to the PIC in complex C resembled that observed for complex B." However this does not appear to be the case the nucleosome in the two complexes appear to engage TFIIH very differently. In complex B there appears to be a lot more contact then in complex C. If this is the case, then it would not support "similar PIC-nucleosome interaction can occur even with different initial PIC-nucleosome distances."

Author Rebuttal, first revision:

Responses to reviewer comments

manuscript NSMB-A46269-T

Structures of transcription preinitiation complex engaged with the +1 nucleosome

Wang, *et al.*

Responses are in italics.

Reviewer #1 (Remarks to the Author):

The revised paper from Wang et al. contains two additional structures that give more context to the structure (now Complex B) from the earlier version. This is helpful in that it shows the +1

nucleosome is not fixed relative to the PIC, validating the concern I raised in my first review. I have no technical issues with the work here, which lives up to the high standards of the Cramer lab. However, I still think the authors' interpretations go beyond what the data really show.

We thank the reviewer for the careful consideration. We have addressed the comments below.

They still claim they've shown "intimate interactions" (abstract) between TFIID and the +1 nucleosome that "stabilize the PIC" (line 199). However, complexes A and B have completely different orientations and contact points (really just proximities, given the low resolutions). Complexes B and C only form after TFIID is already incorporated into the PIC and beginning to translocate. What effect would any PIC stabilization have at this late point? If anything, the inhibition of transcription suggests collision with the nucleosome destabilizes the PIC to ITC conversion by impeding progression to an open complex. The +1 nucleosome might have a positive effect via TFIID bromodomain interactions (which is the mechanism suggested in reference 11), but in the context shown here it clearly has a negative effect.

We observed intimate interactions between TFIID and the +1 nucleosome in complex B. We agree with the reviewer that the +1 nucleosome may not stabilize the PIC directly and have removed this statement in discussion (line 199).

Multiple reviewers suggested that mutagenesis of the proposed TFIID contact points could help test the authors' model, but the authors say those experiments aren't feasible. Without functional validation, the contact points could just be where TFIID and the nucleosome collide as the translocase begins to pull in the DNA. In their reply, the authors seem to be moving in this direction ("we do not think that TFIID specifically recognizes the nucleosome, rather, it serves as a docking site that restricts nucleosome orientation"), but this doesn't come across in the paper, which seems to suggest a stable interaction.

We do not claim that TFIID stably interacts with the nucleosome but suggest that it restricts the rotation of the nucleosome when TFIID and the nucleosome collide. We have clear evidence for this and made this clear in the discussion part.

The observation that Complex C, with ten extra base pairs, shows a similar TFIID-nucleosome orientation as Complex B could be significant, or it might just be because the non-native Widom 601 positioning sequence is only easily unraveled up to this point (as the authors suggest in the last sentences of the discussion). With only the current data, I don't think you can claim this is "a stable intermediate state that gets populated during the initiation phase". It could be one of many transient intermediate rotational positions of the nucleosome that we don't see,

artificially stabilized by the 601 sequence and crosslinking.

We believe that the TFIID-nucleosome orientation we observed in complexes B and C represent an intermediate state since they represent the conformation of the majority of the intact particles in our samples.

Line 201. I don't know of any data suggesting that TFIID activity is affected by histone modifications. Is there any basis for this speculation? Would this be a direct or indirect effect?

We have deleted this speculation.

Is the slight TFIID rotation in Complex B similar to that reported by Murakami in the PIC species leading to ITC (Yang et al., Mol Cell 2022)? If so, that would be worth noting.

This is an interesting point, but no, they are different. We think the slight rotation of TFIID observed here is caused by the collision with the +1 nucleosome.

In sum, I think the work here is technically impressive, but the authors are stuck on a model of a positive TFIID-nucleosome interaction. There's no functional data to support this positive model, and the in vitro transcription inhibition even argues against it. The structures are interesting and worth publishing if the authors were more careful in their interpretation.

We thank the reviewer for the help with improving our manuscript and supporting its timely publication.

Reviewer #2 (Remarks to the Author):

The authors addressed in a satisfactory way all my comments and suggestions. They provide additional structures that confirm their hypothesis and strengthen the manuscript. In my opinion this work is ready to be published in NSMB.

We thank the reviewer again for the suggestions and support.

Reviewer #3 (Remarks to the Author):

In this paper, Wang et al. present the structure of the PIC with an adjacent +1 nucleosome. They determined three structures that are potential intermediates in the collision of the PIC with the +1 nucleosome. The results are interesting and thought provoking, but the authors' many claims need further studies to determine. To dissect this, I will give comment on several of the major claims in the discussion of their paper first. Then go through some the major issues I see in the results.

We thank the reviewer for the insightful comments. We have addressed all the comments and rewritten the discussion.

We report here the first structures of Pol II PIC-nucleosome transcription complexes.
- True, but this is only a partial PIC. Though the reason for this is provided by the author was that TFIID was too flexible in their attempts generate a complex. This could be interesting data to show in the paper.

We agree it would be very interesting to also have included TFIID but for unknown reasons this is technically not feasible in the yeast system so far, despite many years of trials in our (and likely other) laboratories. We note the PIC we have in our structures is sufficient to direct promoter-dependent transcription initiation in the absence of TFIID and thus our structure is functionally relevant.

The structures show that TFIIH interacts with the nucleosome and that a preferred orientation of the nucleosome with respect to the PIC may exist that is characterized by multiple TFIIH-nucleosome contacts.

- First the structure does appear to show multiple contact between TFIIH and the nucleosome however it would seem that each structure appears to show different contacts and as such a "preferred orientation" would not appear to be true.
- Additionally, the contacts they do observe I question the importance of, they are not observed at high enough resolution to show any meaningful interaction (no specific interacting residue pairs can be seen).

We have rewritten this part to make it clearer. Briefly, TFIIH restricts free nucleosome rotation and leads to preferred orientations of the nucleosome when the PIC and nucleosome are in contact distance.

These contacts suggest that the +1 nucleosome can help to stabilize the PIC via TFIIH

interactions and may explain how PIC formation can be influenced by the +1 nucleosome.

- Is the nucleosome-PIC interaction stable? The three states show different interactions. It would seem that the template used for the assembly of these complexes are a bigger factor for how the nucleosome will interact with the PIC.

We have removed this statement since there is no evidence to support that the +1 nucleosome can directly help to stabilize the PIC. Compare also the response to reviewer #1.

The location of TFIID in the vicinity of histone tails might allow the regulation of TFIID activity by histone modifications.

- Is this a known phenomenon? Please give reference. I think this is the first time in the paper this is mentioned.

We have removed this speculative comment. Compare also the response to reviewer #1.

Although the nucleosome may initially be found at different rotational states with respect to the PIC, it is likely that action of the Ssl2 translocase leads to a preferred nucleosome orientation that allows for multiple TFIID contacts as observed here.

- Would it not be possible that if a differently spaced template was used that TFIID could also stabilize this interaction? Complex C used a 10 bp extended template that would give the nucleosome the same rotational state as complex B. Even still it appears that TFIID interact with the nucleosome in a different manner.

This is true. In both complexes B and C, the captured states of the TFIID-nucleosome were after the action of the Ssl2 translocase. Even though that TFIID interacts with the nucleosome in a slightly different manner, the common feature is that TFIID uses the same region to contact the nucleosome, creating a barrier for further rotation of the +1 nucleosome.

The observation that TFIID stops unravelling nucleosomal DNA at SHL -5 indicates that the translocase activity of TFIID is not enough to enable PIC to pass through the strong histone-DNA interacting sites (such as SHL -5 and -1) within nucleosome.

- Is this true in the both the A/B and C template containing structures? The comparison of Complex B and C needs to be expanded in the text and figures. It appears in Figure 5 that the TFIID-nucleosome interactions are different.

Passage of PIC through the +1 nucleosome may require the help from chromatin remodelers and chromatin modifying complexes.

- While not a major point are the authors stating that a chromatin modifier directly helps the passage of the PIC through the +1 or that having a modified nucleosome would help?

- Secondly, would not the acetylation (if this is the type of modification the authors are thinking about) of the +1 typically precede the assembly of the PIC? This would lead to the bigger question as to whether a modified nucleosome would stall the PIC, given that acetylated nucleosome is thought to be slightly less stable than unmodified ones.

As shown in the structures of both complexes B and C (in the presence of NTP), TFIID alone does not allow efficient passage through the +1 nucleosome. We agree with the reviewer that it will be interesting to test whether an acetylated nucleosome will help the passage of PIC but we have no means to test this in a timely manner.

Together with modeling, the structures also provide an explanation for why the transcription start site (TSS) is often located ~60 bp upstream of the dyad of the +1 nucleosome in yeast.

- This is the first sentence of the second and last paragraph of the discussion. This claim is meant to be supported by the rest of this paragraph. I do not see how this claim is at all supported by data.

To initiate RNA chain synthesis, the yeast PIC scans downstream DNA for the TSS.

- While this may seem obvious, the how has never been determined. No one has yet shown the nature of this scanning and how the PIC can start at the TATA box and make its way to the TSS.

- What the authors capture is the beginning of this scanning, and to explain “why the transcription start site (TSS) is often located ~60 bp upstream of the dyad of the +1 nucleosome in yeast” they would need to show the TSS in the active site of the polymerase.

We have tried to explain things better so there is no misunderstandings. The issue is that the barrier to Pol II caused by the nucleosome is well-characterized over the years and in the context of our structures may explain how Pol II progression during scanning is counteracted, and this may lead to usage of a TSS. We think the proposed mechanism is a reasonable model that is consistent with our structures we present. Elucidating the very complicated mechanism of promoter scanning is clearly beyond the scope of this work. But it would be a pity if we are not allowed to speculate on how the nucleosome may influence scanning and the selection of the start site.

Modeling based on complex B indicates that scanning requires further progression of the PIC into the +1 nucleosome and detachment of three additional turns of DNA (Extended Data Fig. 10). This may be achieved by the ATP-dependent Ssl2 translocase that is required for scanning.

- Clearly this is not enough, or else you would see the promoter open.

We speculate that scanning may be impaired at the major nucleosomal barrier just upstream of

the nucleosome dyad, which then may trigger TSS usage and RNA chain initiation as suggested.

- What do the authors mean by “major nucleosomal barrier”? The nucleosome itself or a specific DNA-octamer interaction (e.g. SHL -1)?

This model of nucleosome defined TSS usage may explain why TSSs in yeast are generally located at a distance of ~60 bp from the dyad of the +1 nucleosome, whereas the distance from the TSS to the TATA box varies in the range of ~40-120 bp.

- Yes, the model laid out can explain “why TSSs in yeast are generally located at a distance of ~60 bp from the dyad of the +1 nucleosome,” but it does not explain why “the TSS to the TATA box varies in the range of ~40-120 bp.” Like I stated before the “No one has yet shown the nature of this scanning and how the PIC can start at the TATA box and make its way to the TSS”

In our experimental system, the high stability of the nucleosome on the Widom-601 positioning sequence may have prevented scanning and led to a stable intermediate amenable to structure determination.

- Possibly, and could also have prevented the authors from explaining “why the transcription start site (TSS) is often located ~60 bp upstream of the dyad of the +1 nucleosome in yeast” Future studies should focus on the roles of Mediator and TFIID in nucleosome-involved transcription initiation.

It is indeed possible that some not yet determined factors may be required to help the PIC to scan the promoter for a TSS. We have clarified in the text the use of the term “major nucleosomal barrier” to refer to specific DNA-octamer interactions within the nucleosome (e.g. SHL -1).

While the discussion section needs to be rewritten to better represent the actual finding of this paper. I also have a major concerns with the in vitro transcription assay results and the representation and description of the Complex C state in the results.

Concerns:

In Supplemental Figure 1 (the one that corresponds to Figure 1c) how is the length of the products determined? The gel does not appear to have a ladder. Additionally, FL product is presumably between 80-150bp and would mean that the TSS is locate within the Widom601 sequence. If this is the case, why then is the TSS the same for DNA and Nuc? Is it 60bp form the nucleosome dyad and when there is not nucleosome 60bp after the TATA? And these just happen to be the same place? What would occur with the complex C template? Previously results like from (Nagai et al. PNAS 2017: doi: 10.1073/pnas.1620312114) indicate that given the conditions of this experiment that there should be some product starting at ~30bp form the TATA.

In Supplemental Figure 1 (the one that corresponds to Extended Data Figure 1b) I have the same product size concern as the previous figure. The other concern I have is that the boxed

lanes show a noticeable reduction in the band intensity for the right (Nuc) then any of the other bands. Presumably the other lanes are not just DNA and so which leads mean to wonder which are Nuc and why do they look no way near the intensity of the selected band. The in especially concerning given the supposed deviation of the band intensity is very low based on the circle values shown in Extended Data Figure 1b.

We have made multiple edits to the discussion and hope the reviewer is satisfied. The length of the product was determined based on an RNA ladder on the other gel run under the same conditions. As shown in Fig. 1a, the TSS locates within the Widom 601 sequence. The same sequence was used in in-vitro transcription experiments for naked DNA (DNA) or nucleosome-containing (Nuc) template. Even though there is a potential TSS ~30 bp from the TATA in our template, we didn't observe a clear product starting from this site. A potential explanation for this observation could be that the promoters used in our experiment and their paper are different (His4 vs. Pho5). On the gel that corresponds to Extended Data Figure 1b, the boxed lanes are one of three replicate experiments we carried out on different dates. The other lanes were the same templates tested with other variations (the four lanes to the left) or just DNA (the two lanes to the right). The deviation of the band intensity was calculated between replicates done on different dates. We made sure these things are described correctly in the legend.

Comment:

A more detailed comparison is need for Complexes B and C. Both in the text and in the Figure 4. In the text it is stated "overall orientation of the nucleosome with respect to the PIC in complex C resembled that observed for complex B." However this does not appear to be the case the nucleosome in the two complexes appear to engage TFIIH very differently. In complex B there appears to be a lot more contact then in complex C. If this is the case, then it would not support "similar PIC-nucleosome interaction can occur even with different initial PIC-nucleosome distances."

We have added a more detailed comparison between Complexes B and C in text and Fig. 5. From the new Fig. 5b, it is clear to see a similar orientation between TFIIH and nucleosome in both complex B and C even though the detailed PIC-nucleosome interaction is partially different.

Final Decision Letter:

Message 7th Oct 2022

:

Dear Dr. Cramer,

We are now happy to accept your revised paper "Structures of transcription preinitiation complex engaged with the +1 nucleosome" for publication as a Article in Nature Structural

& Molecular Biology.

Your paper will be published online soon after we receive proof corrections and will appear in print in the next available issue. You can find out your date of online publication by contacting the production team shortly after sending your proof corrections. Content is published online weekly on Mondays and Thursdays, and the embargo is set at 16:00 London time (GMT)/11:00 am US Eastern time (EST) on the day of publication. Now is the time to inform your Public Relations or Press Office about your paper, as they might be interested in promoting its publication. This will allow them time to prepare an accurate and satisfactory press release. Include your manuscript tracking number (NSMB-A46269A) and our journal name, which they will need when they contact our press office.

About one week before your paper is published online, we shall be distributing a press release to news organizations worldwide, which may very well include details of your work.

We are happy for your institution or funding agency to prepare its own press release, but it must mention the embargo date and Nature Structural & Molecular Biology. If you or your Press Office have any enquiries in the meantime, please contact press@nature.com.

Please note that *Nature Structural & Molecular Biology* is a Transformative Journal (TJ). Authors may publish their research with us through the traditional subscription access route or make their paper immediately open access through payment of an article-processing charge (APC). Authors will not be required to make a final decision about access to their article until it has been accepted. [Find out more about Transformative Journals](https://www.springernature.com/gp/open-research/transformative-journals)

Authors may need to take specific actions to achieve [compliance with funder and institutional open access mandates](https://www.springernature.com/gp/open-research/funding/policy-compliance-faqs). If your research is supported by a funder that requires immediate open access (e.g. according to [Plan S principles](https://www.springernature.com/gp/open-research/plan-s-compliance)) then you should select the gold OA route, and we will direct you to the compliant route where possible. For authors selecting the subscription publication route, the journal's standard licensing terms will need to be accepted, including [self-archiving policies](https://www.springernature.com/gp/open-research/policies/journal-policies). Those licensing terms will supersede any other terms that the author or any third party may assert apply to any version of the manuscript.

Kind regards,
Florian

Dr Florian Ullrich
Associate Editor, Nature
Consulting Editor, Nature Structural & Molecular Biology
ORCID 0000-0002-1153-2040